# ROVER: Benchmarking Reciprocal Cross-Modal Reasoning for Omnimodal Generation

**Yongyuan Liang**[*△], **Wei Chow**[*♠], **Feng Li**[◇], **Ziqiao Ma**[♣], **Xiyao Wang**[△],
**Jiageng Mao**[★], **Jiuhai Chen**[△], **Jiatao Gu**[♠], **Yue Wang**[†★] **Furong Huang**[†△]
[△]University of Maryland, College Park    [♠]University of Pennsylvania
[♣]University of Michigan    [★]University of Southern California
[◇]The Hong Kong University of Science and Technology

## ABSTRACT

Unified multimodal models (UMMs) have emerged as a powerful paradigm for seamlessly unifying text and image understanding and generation. However, prevailing evaluations treat these abilities in isolation, such that tasks with multimodal inputs and outputs are scored primarily through unimodal reasoning, i.e., textual benchmarks emphasize language-based reasoning, while visual benchmarks emphasize reasoning outcomes manifested in the pixels. We introduce ROVER to address this pressing need to test *reciprocal cross-modal reasoning*, the use of one modality to guide, verify, or refine outputs in the other, an ability central to the vision of unified multimodal intelligence. ROVER is a human-annotated benchmark that explicitly targets reciprocal cross-modal reasoning, which contains 1,312 tasks grounded in 1,876 images, spanning two complementary settings. **Verbally-augmented reasoning for visual generation** evaluates whether models can use verbal prompts and reasoning chains to guide faithful image synthesis. **Visually-augmented reasoning for verbal generation** evaluates whether models can generate intermediate visualizations that strengthen their own reasoning processes for question answering. Experiments on 17 unified models reveal two key findings: (i) Cross-modal reasoning determines visual generation quality, with interleaved models significantly outperforming non-interleaved ones; notably, combining strong unimodal models fails to achieve comparable reasoning. (ii) Models show dissociation between physical and symbolic reasoning: they succeed at interpreting perceptual concepts literally but fail to construct visual abstractions for symbolic tasks, where faulty reasoning harms performance. These results highlight reciprocal cross-modal reasoning as a critical frontier for enabling true omnimodal generation. Homepage: *roverbench.github.io*.

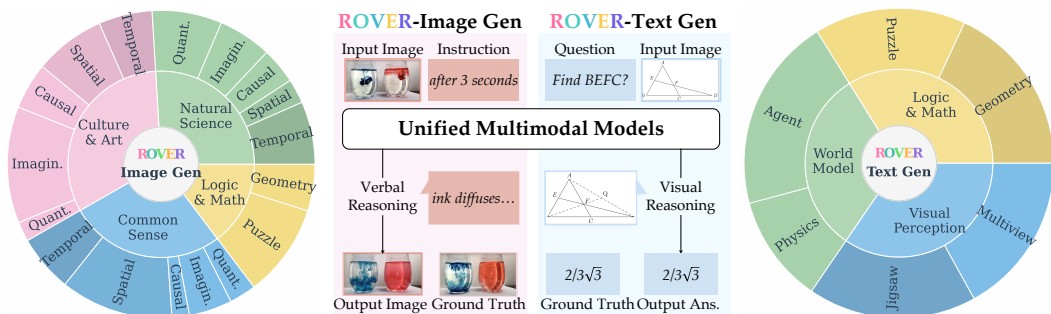

Figure 1: The ROVER benchmark. ROVER evaluates UMMs through reciprocal cross-modal reasoning: **ROVER-IG** (left) requires generating images with language-augmented reasoning, while **ROVER-TG** (right) requires generating text answers with visually-augmented reasoning.

---

* Authors contributed equally to this work. † Advisors contributed equally to this work.

# 1 INTRODUCTION

The development of *unified multimodal models* (also referred to as *omnimodal models*) has drawn significant attention to their potential for unified understanding and generation across text and images (Comanici et al., 2025; Hurst et al., 2024; Tong et al., 2024; Deng et al., 2025; Xu et al., 2025b). However, prevailing evaluations treat these abilities in isolation, such that tasks with multimodal inputs and outputs are scored primarily through unimodal reasoning: textual benchmarks emphasize language-based reasoning, while visual benchmarks emphasize reasoning outcomes manifested in the pixels. On the language side, evaluation focuses on generating text in response to an image and an accompanying question, thereby testing perceptual understanding (Chen et al., 2024; Liu et al., 2024; Yu et al., 2024) and reasoning (Lu et al., 2023; Yue et al., 2024; Wang et al., 2024; Hao et al., 2025; Gao et al., 2025). On the vision side, evaluation centers on generating images conditioned on either instructions or text-image pairs, thereby testing direct image generation (Ghosh et al., 2023; Ma et al., 2024; Niu et al., 2025) or image editing (Kawar et al., 2023; Zhang et al., 2023; Ma et al., 2024; Sheynin et al., 2024; Yu et al., 2025a; Liu et al., 2025b; Wu et al., 2025e).

Unlike earlier multimodal systems that specialize in either visual perception or generation, UMMs are designed to reason seamlessly across modalities and produce outputs that span both. This creates a pressing need for benchmarks that evaluate their ability to use one modality to guide, verify, or refine outputs in the other. We refer to this capability as *reciprocal cross-modal reasoning* (Figure 1). To benchmark such capability in current unified multimodal models, we present **ROVER**, a human-annotated and rigorously verified benchmark with over 1,312 tasks grounded in 1,876 images. **ROVER** targets two complementary settings: (i) **verbally-augmented reasoning for visual generation**, including 4 conceptual domains (natural science, culture & art, common sense, and logic & math) with high complexity are instantiated across 7 reasoning subtasks: temporal, spatial, causal, synthetic, quantitative, abstract, and mathematical. Each instance provides a textual prompt with an initial image and a chain of constraints that a correct output image must satisfy. (ii) **visually-augmented reasoning for verbal generation**, including 6 subtask variants spanning 3 problem domains: visual perception, world modeling for robot manipulation & physical dynamics prediction, and logical reasoning for geometry & puzzle solving. Instances interleave turns of text and images, requiring the model to emit visual intermediates that make downstream reasoning auditable.

A key challenge is that evaluating reciprocal cross-modal reasoning requires assessing both the rationales and the output. Text-only metrics overlook visual fidelity, while image-only metrics cannot verify whether the image reflects valid reasoning. Human evaluation provides accurate judgments but is prohibitively expensive at scale. To address this, we adopt a multi-dimensional protocol that combines an automated VLM judge (based on GPT-4.1 (Hurst et al., 2024)) with expert validation on stratified samples. The judge is supplied with rubric cards and reference assets and scores along three reasoning-specific dimensions: (i) the logical coherence of domain-specific reasoning processes, (ii) the alignment of generated outputs with target descriptions or ground-truth answers, and (iii) the consistency between intermediate reasoning steps and the final images or answers. For visual generation tasks, the framework additionally incorporates established image consistency and quality metrics (Hu et al., 2023; Wu et al., 2023; Kirstain et al., 2023; Xu et al., 2023; Brooks et al., 2023). The judge is calibrated with expert explanations, and its agreement with expert evaluations is reported, following recent LLM-as-judge methodologies (Kim et al., 2023; Hu et al., 2023).

Through extensive evaluation of 17 unified multimodal models, our experiments reveal significant gaps in cross-modal reasoning capabilities. Specifically, unified models exhibit: (1) Cross-modal reasoning capabilities largely affect reasoning-dependent visual generation quality, with interleaved image-text generation models significantly outperforming non-interleaved counterparts; critically, strong unimodal models cannot replicate this cross-modal reasoning on **ROVER**, even when combined; (2) Striking dissociation between physical and symbolic visual reasoning. While unified models excel at generating visual reasoning steps for perceptual and physical world concepts through literal interpretation of visual elements, they fundamentally struggle to reason about visual abstractions as symbolic representations (Hsu et al., 2025), causing poor visual reasoning to degrade downstream performance rather than improving it. These findings reveal fundamental capability limitations in current unified models, underscoring the critical role of reciprocal cross-modal reasoning for holistic omnimodal generation, where independent optimization of constituent modalities proves insufficient.

Table 1: **Summary of Multimodal Reasoning Benchmarks.** We compare existing works from aspects including: [1]interleave: supports multi-image or multi-turn inputs; [2]process evaluation: evaluates intermediate reasoning steps; [3]vision necessity: requires reasoning grounded in visual understanding; [4]multidimensional evaluation: scores models along multiple dimensions; [5]hybrid evaluation: uses GPT-based judgments instead of purely visual metrics; [6] manual annotations: whether manual annotations and filtering are applied; [7]scale: dataset scale; [8]types: data categories.

| Benchmark | Venue | Inter. | Process Eval | Vision Necess. | Multi. | Hybrid Eval | Manual Anno. | #Types |
|---|---|---|---|---|---|---|---|---|
| ReasonPix2Pix (Jin et al., 2024) | arXiv'24 | ✗ | ✗ | ✗ | ✗ | ✗ | ✗ | 1 |
| ReasonEdit (Huang et al., 2024) | CVPR'24 | ✗ | ✗ | ✗ | ✗ | ✗ | ✓ | 1 |
| EditWorld (Yang et al., 2024) | MM'25 | ✗ | ✗ | ✗ | ✗ | ✗ | ✗ | 7 |
| Reason50K (He et al., 2025) | arXiv'25 | ✗ | ✗ | ✗ | ✗ | ✗ | ✗ | 4 |
| KRIS-Bench (Wu et al., 2025e) | NeurIPS'25 | ✗ | ✗ | ✗ | ✓ | ✓ | ✓ | 7 |
| RISEBench (Zhao et al., 2025) | NeurIPS'25 | ✗ | ✗ | ✓ | ✓ | ✓ | ✓ | 4 |
| WorldGenBench (Zhang et al., 2025) | arXiv'25 | ✗ | ✗ | ✗ | ✓ | ✓ | ✗ | 2 |
| Unified-Bench (Yan et al., 2025) | arXiv'25 | ✗ | ✗ | ✗ | ✗ | ✗ | ✗ | 1 |
| MetaQuery (Pan et al., 2025) | arXiv'25 | ✓ | ✗ | ✓ | ✗ | ✗ | ✗ | - |
| **ROVER** | **Ours** | ✓ | ✓ | ✓ | ✓ | ✓ | ✓ | 23 |

Our main contributions are summarized as follows:

- We introduce **ROVER**, the first benchmark that explicitly targets **reciprocal** cross-modal reasoning for visual generation and interleaved multimodal reasoning.
- We provide a principled task taxonomy and a verification-ready instance design with process targets and visual artifacts, together with a multi dimensional protocol that scores coherence, alignment, and step to output consistency.
- We evaluate 17 unified models, uncovering significant limitations in cross-modal reasoning and providing the community with insights on unified model development toward truly omnimodal generation.

## 2 RELATED WORK

**Unified Multimodal Models (UMMs).** UMMs represent a paradigm of architectures designed to seamlessly integrate multimodal comprehension and generation capabilities within a singular, cohesive framework. To achieve this unified objective, seminal works (Karypis et al., 1999; Wu et al., 2025b; Chen et al., 2025b) leverage image tokenization strategies, employing autoregressive next-token prediction paradigms to generate visual tokens. Building upon these foundations, Show-o2 (Xie et al., 2025) introduces discrete diffusion scheduling mechanisms to enhance the token prediction process and improve generation fidelity. Subsequent developments, driven by the pursuit of enhanced image synthesis quality, incorporate diffusion-based or flow-matching heads (Lipman et al., 2022) integrated with shared transformer architectures (Deng et al., 2025; Ma et al., 2025; Zhou et al., 2024). Alternative approaches within the UMM paradigm maintain powerful pretrained backbone in a frozen state for reasoning tasks, while routing their intermediate feature representations through learnable query mechanisms to external image generation modules (Pan et al., 2025; Wu et al., 2025d). However, the comprehensive evaluation of synergistic relationships between multimodal understanding, reasoning, and generation in UMMs remains largely unexplored, with existing benchmarks inadequately assessing whether these capabilities exhibit mutual enhancement or coordination deficiencies.

**Reasoning-Guided Image Generation.** With the emergence of UMMs, multimodal reasoning has garnered increasing attention from the research community. However, the majority of existing work remains focused on instruction comprehension, namely leveraging input images to perform instruction translation and subsequently generate corresponding visual outputs (Jin et al., 2024; Huang et al., 2024; Yang et al., 2024; He et al., 2025; Wu et al., 2025e; Yu et al., 2025b). Unified-Bench (Yan et al., 2025) employs iterative image-text generation to measure the degree of unification between comprehension and generation models. RISEBench (Zhao et al., 2025) extends beyond prior work by introducing LMM-as-a-judge to evaluate visual rationality in addition to assessing image consistency, yet remains limited to computing similarity scores against human-provided ground truth. However, these benchmarks lack comprehensive evaluation beyond image consistency, particularly overlooking the intermediate reasoning processes, such as whether the rationale is sound and aligns with the generation outcome. In contrast, **ROVER** represents the first benchmark to in-

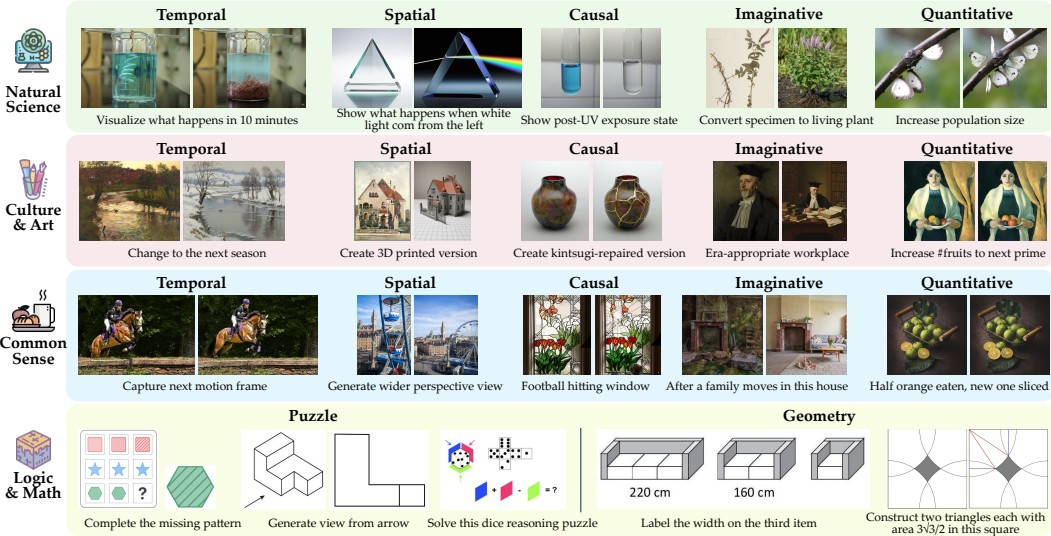

Figure 2: Overview of **ROVER-IG**, the benchmark for evaluating how unified multimodal models generate images under intensive verbal reasoning. The benchmark spans 4 domains (natural science, culture and art, common sense, and logic), instantiated across 7 reasoning subtasks.

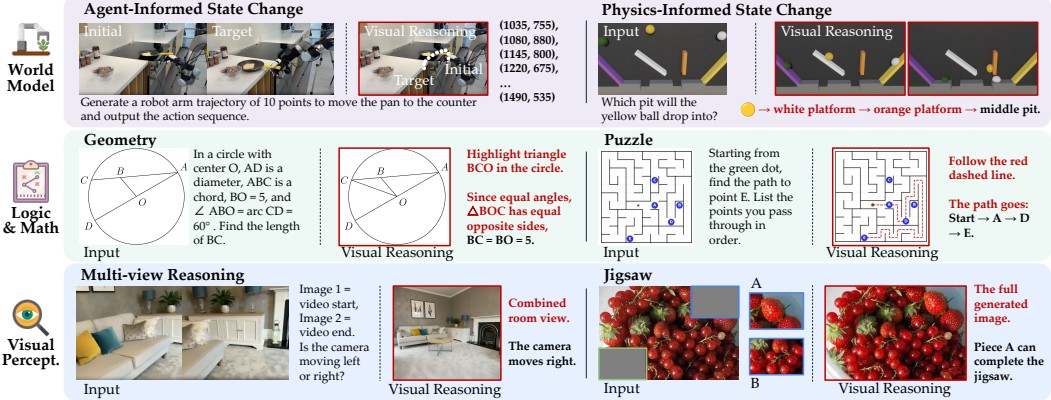

Figure 3: Overview of **ROVER-TG**, the benchmark for evaluating visually-augmented reasoning in verbal generation. The benchmark spans 3 scenarios and 6 subtasks: physical world modeling, logical assistance, and visual perception enhancement.

vestigate the interplay between reasoning and generation. A detailed comparison can be found in Table 1. A more detailed discussion about interleaved reasoning can be found in Appendix C.

# 3 ROVER BENCHMARK

## 3.1 VERBALLY-AUGMENTED REASONING FOR VISUAL GENERATION

We introduce **ROVER-IG**, a benchmark designed to evaluate how UMMs generate images when jointly guided not only by visual understanding but also by intensive language reasoning. **Taxonomy.** It spans 4 domains and 7 reasoning subtasks, each demanding complex text-driven reasoning chains to direct image generation and test models' ability to integrate text-augmented reasoning with visual synthesis. Figure 2 provides a visual overview of our benchmark taxonomy and representative examples.

- **Domains.** We categorize tasks across 4 distinct areas: **Nature Science** encompasses scientific phenomena, experimental processes, and fundamental laws of nature; **Culture & Art** includes artistic creation, cultural artifacts, humanities, and aesthetic principles; **Common Sense** covers

everyday scenarios requiring intuitive understanding and practical reasoning; **Logic & Math** focuses on abstract visual puzzles, mathematical relationships, and general pattern discovery.

- **Reasoning subtasks.** We define 5 core reasoning capabilities: **Temporal** involves sequence prediction, progression analysis, and time-based changes; **Spatial** requires understanding geometric relationships, perspective changes, and spatial visualization; **Causal** connects cause-effect relationships and mechanism understanding; **Imaginative** combines multiple elements through creative integration and novel object generation; **Quantitative** involves numerical changes, scaling operations, and mathematical relationships. The Logic domain additionally includes two specialized reasoning types: **Puzzle** for abstract visual pattern discovery and problem solving, and **Geometry** for geometrical principles applied to visual generation.

**Data Curation.** We curated our dataset through a systematic multistage process, beginning with human experts selecting candidate images from large-scale web image datasets. For each selected image, domain experts and large language models collaboratively generated reasoning tasks that require genuine visual understanding and complex reasoning chains. Each task includes 4 key components: the reasoning prompt specifying the required generation results, target descriptions detailing expected visual outcomes, domain-specific keywords identifying relevant concepts that should guide the reasoning process, and optionally target reference images for validation purposes. All generated tasks were subjected to final human verification to confirm the complexity and rationality of the reasoning. Our final dataset comprises 908 visual generation tasks involving 1,009 images, with both single-image and multi-image generation scenarios distributed across all reasoning subtasks and domains.

**Evaluation Metrics.** Ideally, the evaluation protocol should cover both the reasoning process and the resulting outputs. As human evaluation is prohibitively costly at scale, we automated the evaluation following LMM-as-judge. We assess model performance across 5 rubric dimensions designed to capture the effectiveness of reasoning-to-generation workflows. **Reasoning Process (RP)** evaluates the quality of verbal reasoning through logical structure, domain knowledge application, reasoning type-specific validation, and completeness assessment. **Reasoning Visual (RV)** measures how well the generated visual output matches target descriptions and demonstrates correct reasoning principles. **Reasoning Alignment (Align.)** specifically quantifies the consistency between verbal reasoning processes and visual generation outcomes, addressing whether models can effectively translate reasoning into visual results. **Visual Consistency (VC)** ensures that non-target elements remain unchanged during reasoning-guided generation, validating precise control capabilities. **Image Quality (IQ)** assesses the technical excellence and visual coherence of generated images, including structural coherence, visual fidelity, and absence of generation artifacts.

### 3.2 VISUALLY-AUGMENTED REASONING FOR VERBAL GENERATION

We then introduce **ROVER-TG**, the benchmark counterpart for evaluating how UMMs generate language responses guided by interleaved reasoning with visually-augmented rationale. Unlike text-only Chain-of-Thought, we examine scenarios where models generate intermediate visual representations to facilitate reasoning. This interleaved reasoning paradigm reflects human cognitive patterns that integrate verbal and visual thinking for complex problem solving (Barsalou, 1999).

**Taxonomy.** We focus on 3 scenarios, comprising 6 challenge types and 404 tasks, where visual generation genuinely enhances reasoning beyond text-only rationale, as shown in Figure 3: physical world simulation, logical problem solving with visual aids, and enhanced visual perception through generated representations.

- **World Model.** Tasks require models to act as world simulators, predicting intermediate visual states that reflect environment dynamics under given initial conditions and actions. Models must generate these states from robotic actions or physical processes and leverage them for embodied planning, spatial reasoning, and motion prediction.

- **Logic & Math.** Tasks involve generating visual aids to solve abstract puzzles and geometry problems, similar to how humans draw auxiliary lines, diagrams, or visual representations to facilitate logical reasoning. Models must create helpful visual elements that make implicit relationships explicit and support step-by-step logical inference processes.

- **Visual Perception.** Tasks focus on generating supportive images to improve performance on challenging visual perception problems, including multi-view reasoning and jigsaw puzzles. The

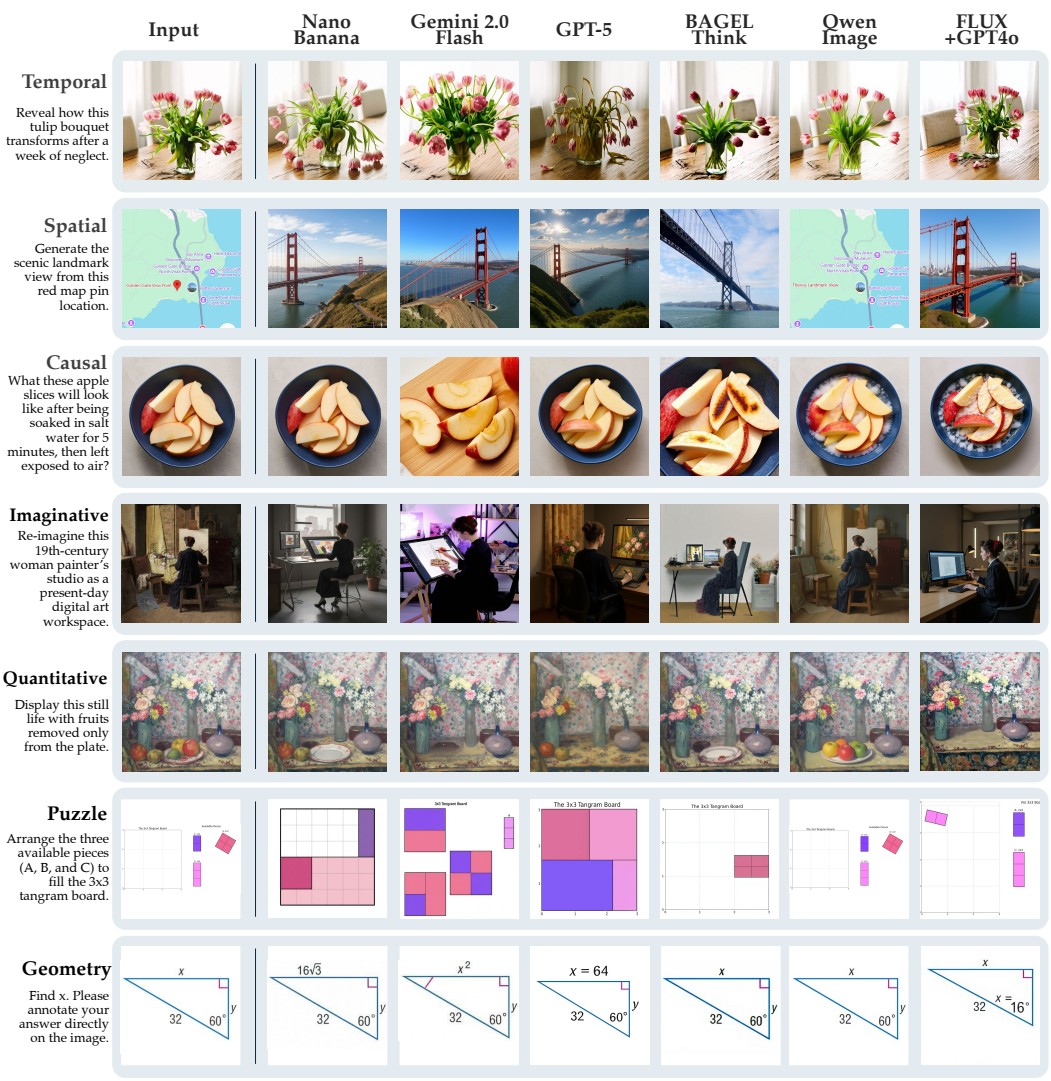

Figure 4: Example outputs on **ROVER-IG**. Each row corresponds to one reasoning subtask, with the input on the left and outputs from representative unified multimodal models shown across columns. Verbal reasoning outputs are shown in Figure 11.

generated images in the rationale serve as intermediate representations that reduce hallucinations and improve accuracy in visual understanding tasks.

**Data Curation.** Our dataset compilation draws from diverse sources including robotics datasets, physical simulation videos, logic puzzles, and challenging perception tasks. We establish a consistent structure for each task: contextual setup with initial images, progressive reasoning steps, and verified ground truth solutions. Crucially, our curation process ensures that generated visuals serve as active reasoning components rather than decorative elements, thereby fully leveraging omnimodal generation capabilities to tackle complex problem-solving scenarios.

**Evaluation Metrics.** Similarly, we automated the evaluation using a VLM judge across 3 rubric dimensions. **Interleaved Reasoning Quality (IR)** evaluates the plausibility and relevance of intermediate visual representations through physical/logical correctness, task-specific utility, visual coherence, and reasoning completeness. **Final Answer Accuracy (Acc.)** measures whether the model's final reasoning outcome matches the provided ground truth answer across all three scenario types. **Reasoning-Answer Alignment (Align.)** quantifies how effectively generated images contribute to reaching correct conclusions, examining causal relationships between visual aids and final

Table 2: **Main Results on Verbally-Augmented Visual Generation.** We evaluate 13 closed- and open-source unified models across four conceptual domains. **Verb.-Aug.** denotes verbally-augmented reasoning. Performance is measured using three key metrics: **Reasoning Process (RP)**, which assesses the logical quality of the verbal reasoning; **Alignment (Align.)**, which quantifies the consistency between the reasoning process and the generated visual output; and **Reasoning Visual (RV)**, which measures how well the final image reflects the target description.

| Verb.-Aug. Reasoning for Visual Generation | Nature Science | | | Culture & Art | | | Common Sense | | | Logic & Math | | | Overall | | |
|---|---|---|---|---|---|---|---|---|---|---|---|---|---|---|---|
| | RP | Align. | RV | RP | Align. | RV | RP | Align. | RV | RP | Align. | RV | RP | Align. | RV |
| **Closed-source Unified Models** | | | | | | | | | | | | | | | |
| Nano Banana (Comanici et al., 2025) | 64.8 | 88.8 | 77.3 | 68.1 | 81.9 | 76.6 | 61.8 | 85.0 | 74.8 | 78.6 | 66.1 | 55.1 | **67.0** | **82.3** | **73.2** |
| Gemini 2.0 Flash (Comanici et al., 2025) | 64.1 | 88.4 | 68.8 | 62.8 | 78.7 | 71.9 | 57.8 | 74.4 | 66.1 | 74.5 | 63.2 | 42.6 | 64.8 | 78.6 | 62.3 |
| GPT-5 (Hurst et al., 2024) | 61.7 | 87.9 | 71.3 | 63.4 | 80.2 | 72.6 | 56.3 | 77.2 | 65.3 | 75.4 | 60.2 | 45.8 | 64.2 | 76.4 | 63.7 |
| **Open-source Unified Models** | | | | | | | | | | | | | | | |
| BAGEL-Think (Deng et al., 2025) | 58.1 | 64.2 | 54.0 | 53.2 | 78.0 | 63.7 | 50.1 | 69.4 | 55.9 | 57.7 | 26.2 | 20.8 | **54.3** | **64.4** | **52.7** |
| BAGEL (Deng et al., 2025) | - | - | 35.9 | - | - | 49.2 | - | - | 42.0 | - | - | 27.1 | - | - | 40.5 |
| Step1X-Edit v1.2 (Liu et al., 2025a) | 29.7 | 59.7 | 46.2 | 31.4 | 71.6 | 50.6 | 28.7 | 61.0 | 46.1 | 77.5 | 35.5 | 18.4 | 37.0 | 60.3 | 43.5 |
| UniCoT (Qin et al., 2025) | 52.4 | 68.9 | 38.2 | 57.3 | 69.2 | 63.9 | 53.1 | 64.3 | 56.3 | 50.3 | 23.1 | 21.5 | 50.7 | 56.3 | 47.4 |
| BLIP3o-NEXT (Chen et al., 2025a) | - | - | 38.2 | - | - | 47.5 | - | - | 43.3 | - | - | 22.5 | - | - | 37.8 |
| Ovis-U1 (Wang et al., 2025) | - | - | 28.6 | - | - | 44.3 | - | - | 42.1 | - | - | 20.5 | - | - | 33.8 |
| UniPic2-Metaquery-9B (Wei et al., 2025) | - | - | 33.8 | - | - | 52.7 | - | - | 43.2 | - | - | 27.1 | - | - | 39.2 |
| ILLUME+ (Huang et al., 2025) | - | - | 28.1 | - | - | 43.2 | - | - | 36.9 | - | - | 20.1 | - | - | 32.0 |
| Emu2-Gen (Sun et al., 2023) | - | - | 29.1 | - | - | 42.6 | - | - | 37.4 | - | - | 20.3 | - | - | 32.3 |
| OmniGen2 (Wu et al., 2025c) | - | - | 27.4 | - | - | 42.3 | - | - | 39.2 | - | - | 20.2 | - | - | 32.2 |

outputs, reasoning chain coherence, and whether the visual generation process was necessary for successful task completion.

# 4 EXPERIMENTS

## 4.1 EVALUATION SETUP

**Models.** We evaluate a diverse set of models across different categories. For closed-source unified models, we assess three state-of-the-art systems: Gemini 2.5 Flash Image (a.k.a Nano Banana) (Comanici et al., 2025), Gemini 2.0 Flash (Comanici et al., 2025), and GPT-5 (Hurst et al., 2024). For open-source unified models, we evaluate ten representative models including BAGEL-Think and BAGEL (Deng et al., 2025), UniCoT (Qin et al., 2025), Step1X-Edit v1.1/v1.2 (Liu et al., 2025b), BLIP3o-NEXT (Chen et al., 2025a), Ovis-U1 (Wang et al., 2025), UniPic2-Metaquery-9B (Wei et al., 2025), ILLUME+ (Huang et al., 2025), Emu2-Gen (Sheynin et al., 2024), OmniGen2 (Wu et al., 2025c). We also compare against specialized image editing models, including Qwen-Image-Edit (Wu et al., 2025a), FLUX.1 Kontext (Labs et al., 2025), UltraEdit (SD3) (Zhao et al., 2024), VAREedit-8B (Mao et al., 2025). Additionally, we include reasoning language models such as GPT-4.1 (Hurst et al., 2024) to present verbal-only reasoning baselines. All evaluation details are provided in Appendix F.

**Evaluation Protocol.** We employ GPT-4.1 as the automatic judge to assess model outputs across multiple dimensions. All metrics are scored on a 5-point scale (1-5) and normalized to a 0-100 scale for consistent comparison. For VQA problems in **ROVER-TG** with objective answers, **Acc.** denotes exact answer accuracy.

## 4.2 VERBALLY-AUGMENTED REASONING FOR VISUAL GENERATION

**Cross-modal reasoning capabilities and alignment strongly correlate with visual generation effectiveness.** The consistent pattern across all models and dimensions in Table 2. Closed-source models excel in reasoning processes and demonstrate strong alignment performance, which directly contributes to their superior visual generation quality. In contrast, open-source models show notably weaker verbal reasoning during visual generation tasks—their reasoning processes are approximately 38% lower and alignment performance falls about 31% short of closed-source models. This substantial reasoning gap translates into correspondingly diminished visual generation performance that is approximately 39% lower than closed-source models. This finding confirms that cross-modal reasoning capabilities serve as the fundamental driver of visual generation effectiveness on **ROVER-IG**, with stronger reasoning processes and better alignment consistently enabling superior visual output quality.

Table 3: **Performance on visually-augmented reasoning.** We evaluate 6 leading unified and language models across three problem types, comparing two distinct reasoning modes. **Verb.** denotes standard verbal reasoning, where the model generates a final answer directly from the prompt. **Verb.+Vis.** denotes visually-augmented reasoning, where the model generates intermediate visual artifacts to support its final answer. We report on the quality of **Interleaved Reasoning (IR)**, **Alignment (Align.)**, and **Final Answer Accuracy (Acc.)**.

| Verb.+Vis. Reasoning for Verbal Generation | Reasoning Modalities | World Model | | | Logic & Math | | | Visual Perception | | | Overall | | |
|---|---|---|---|---|---|---|---|---|---|---|---|---|---|
| | | IR | Align. | Acc. | IR | Align. | Acc | IR | Align. | Acc. | IR | Align. | Acc. |
| **Closed-source Unified Models** | | | | | | | | | | | | | |
| Nano Banana (Comanici et al., 2025) | Verb.+Vis. | 35.3 | 62.0 | 40.6 | 14.8 | 61.2 | 44.9 | 66.5 | 56.8 | 50.0 | **38.8** | **60.0** | **43.6** |
| | Verb. | - | - | 36.9 | - | - | 42.0 | - | - | 43.7 | - | - | 40.8 |
| Gemini 2.0 Flash (Comanici et al., 2025) | Verb.+Vis. | 27.1 | 46.7 | 35.6 | 11.4 | 47.9 | 30.4 | 49.5 | 46.8 | 43.0 | 29.3 | 47.1 | 36.3 |
| | Verb. | - | - | 33.2 | - | - | 32.6 | - | - | 39.8 | - | - | 35.2 |
| GPT-5 (Hurst et al., 2024) | Verb.+Vis. | 32.8 | 61.5 | 39.2 | 13.2 | 58.7 | 45.6 | 62.7 | 54.9 | 45.5 | 36.2 | 60.9 | 43.4 |
| | Verb. | - | - | 39.2 | - | - | 45.6 | - | - | 43.6 | - | - | 42.8 |
| **Open-source Unified Models** | | | | | | | | | | | | | |
| BAGEL-Think (Deng et al., 2025) | Verb.+Vis. | 22.3 | 34.7 | 26.6 | 10.8 | 36.9 | 24.6 | 31.2 | 44.3 | 34.1 | 21.4 | 38.6 | **28.4** |
| | Verb. | - | - | 24.9 | - | - | 23.1 | - | - | 32.2 | - | - | 26.7 |
| UniCoT (Liu et al., 2025b) | Verb.+Vis. | 22.1 | 35.4 | 26.7 | 10.6 | 38.8 | 21.7 | 34.2 | 42.3 | 34.1 | **22.3** | **38.8** | 27.5 |
| | Verb. | - | - | 24.6 | - | - | 21.7 | - | - | 33.5 | - | - | 26.7 |
| Qwen-2.5-VL-7B (Bai et al., 2025) | Verb. | - | - | 24.2 | - | - | 22.4 | - | - | 32.9 | - | - | 26.5 |
| **Reasoning Language Models** | | | | | | | | | | | | | |
| GPT-4.1 (Liu et al., 2025b) | Verb. | - | - | 37.8 | - | - | 31.8 | - | - | 37.9 | - | - | 35.8 |

**Models capable of interleaved image-text generation demonstrate superior visual generation performance.** Our results reveal a significant performance gap between models that support interleaved generation and those limited to single-turn, single-modality outputs. Among the open-source models evaluated, those with interleaved generation capabilities demonstrate markedly superior performance on Reasoning Visual (**RV**) metric—approximately 38.1% higher than non-interleaved models. This performance advantage suggests that reasoning and generation processes are synergistic, effectively enhancing the model's performance in visual expression tasks.

**Unified models demonstrate absolute advantages over image editing models across visual quality metrics on reasoning-dependent tasks.** As shown in Table 4, Unified models substantially outperform specialized image editing models across all visual quality metrics on **ROVER-IG**. While existing editing models excel at complex text rendering and precise image editing consistency, they fundamentally lack the internal reasoning capabilities required for our reasoning-dependent visual generation tasks. This performance gap fully demonstrates that ROVER effectively evaluates cross-modal reasoning capabilities essential for visual generation.

## 4.3 VISUALLY-AUGMENTED REASONING FOR VERBAL GENERATION

**Current unified models exhibit limited capacity in visual reasoning, constraining their ability to leverage visual augmentation for improved performance.** The evaluation in Table 3 reveals fundamental limitations in generating meaningful visual reasoning steps, with even the best-performing models achieving only 38.8% average **IR** quality and open-source models lagging substantially behind. Models with weaker visual reasoning capabilities show minimal or even negative improvements in final accuracy compared to pure text-based reasoning. Flawed visual reasoning proves worse than no visual reasoning at all.

**Visual reasoning proves effective for physical world tasks but fails systematically on symbolic reasoning.** Across all models, visual augmentation consistently improves performance on World Model and Visual Perception tasks, where visual reasoning steps naturally align with physical phenomena. In stark contrast, Logic & Math tasks reveal systematic failures with minimal and unstable improvements, exposing a fundamental inability to visually symbolize abstract reasoning. Models struggle to create visual representations that capture symbolic reasoning processes (e.g., auxiliary lines in geometry, Figure 5), as the mapping from symbolic logic to visual form remains poorly developed.

## 4.4 FURTHER ANALYSES AND DISCUSSIONS

**Cross-modal reasoning matters for UMMs.** To validate that UMMs perform cross-modal reasoning internally and that this mechanism cannot be replicated through external models serving

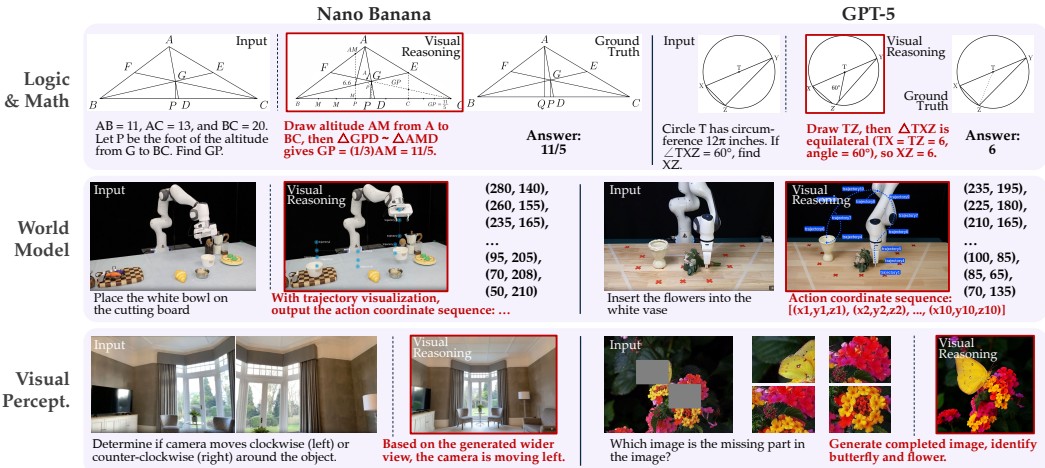

Figure 5: Example outputs on **ROVER-TG**. Each row corresponds to one reasoning scenario, with the input on the left and outputs from representative unified models shown across columns.

Table 4: Visual performance comparison with image editing models on **ROVER-IG** benchmark. We evaluate image editing models and unified models, measuring **Reasoning Visual (RV)**, **Visual Consistency (VC)**, and **Image Quality (IQ)** performance.

| Visual Generation Quality | Nature Science | | | Culture & Art | | | Common Sense | | | Logic & Math | | | Overall |
|---|---|---|---|---|---|---|---|---|---|---|---|---|---|
| | RV | VC | IQ | RV | VC | IQ | RV | VC | IQ | RV | VC | IQ | |
| **Image Editing Models** | | | | | | | | | | | | | |
| Qwen-Image-Edit (Wu et al., 2025a) | 46.7 | 69.1 | 89.8 | 62.5 | 69.6 | 95.2 | 53.1 | 74.2 | 94.4 | 30.4 | 64.5 | 87.2 | 47.1 |
| FLUX.1 Kontext (Labs et al., 2025) | 37.4 | 61.9 | 83.5 | 44.9 | 64.6 | 88.8 | 42.3 | 62.1 | 85.0 | 20.2 | 50.6 | 78.2 | 40.9 |
| UltraEdit(SD3) (Zhao et al., 2024) | 27.0 | 43.6 | 75.7 | 45.2 | 42.6 | 79.0 | 27.9 | 37.3 | 74.7 | 25.2 | 60.1 | 76.1 | 34.6 |
| VAREdit-8B (Mao et al., 2025) | 34.6 | 64.3 | 75.4 | 46.5 | 58.5 | 78.2 | 33.6 | 59.0 | 75.0 | 17.4 | 46.6 | 57.1 | 37.5 |
| Step1X-Edit v1.1 (Liu et al., 2025a) | 38.2 | 75.7 | 85.5 | 50.5 | 62.7 | 83.8 | 35.2 | 67.9 | 85.3 | 16.1 | 61.1 | 85.9 | 42.1 |
| Step1X-Edit v1.2 (Liu et al., 2025a) | 46.2 | 76.8 | 80.6 | 50.6 | 63.0 | 79.2 | 46.1 | 67.2 | 79.6 | 18.4 | 61.1 | 72.2 | **57.4** |
| **Closed-source Unified Models** | | | | | | | | | | | | | |
| Nano Banana (Comanici et al., 2025) | 77.3 | 85.7 | 87.0 | 76.6 | 78.4 | 89.2 | 74.8 | 87.1 | 93.8 | 55.1 | 70.3 | 81.0 | **79.6** |
| Gemini 2.0 Flash (Comanici et al., 2025) | 68.8 | 72.0 | 81.1 | 71.9 | 65.3 | 83.2 | 66.1 | 76.4 | 91.2 | 42.6 | 68.0 | 79.3 | 72.1 |
| GPT-5 (Hurst et al., 2024) | 71.3 | 69.9 | 90.5 | 72.6 | 58.8 | 96.0 | 65.3 | 80.9 | 87.2 | 45.8 | 74.9 | 86.6 | 74.9 |
| **Open-source Unified Models** | | | | | | | | | | | | | |
| BAGEL-Think (Deng et al., 2025) | 54.0 | 65.5 | 78.0 | 63.7 | 65.8 | 71.6 | 55.9 | 76.9 | 80.2 | 20.8 | 48.7 | 76.6 | **62.9** |
| BAGEL (Deng et al., 2025) | 35.9 | 53.6 | 69.9 | 49.2 | 50.2 | 71.9 | 42.0 | 59.1 | 73.0 | 27.1 | 59.2 | 79.8 | 37.8 |

as intermediate reasoning agents, we conduct a comparative analysis in Figure 6 between BAGEL (UMM), FLUX.1 Kontext (Labs et al., 2025), and its GPT-4o–refined cascade variant (FLUX+GPT). Key findings are: (1) *UMMs enable superior cross-modal reasoning.* The think mechanism consistently improves performance on ROVER, boosting visual consistency by 11.9%. Results on Edit-World, where lower CLIP-I indicates more substantive edits, show that external textual refinement can benefit editing tasks but does not translate to the cross-modal reasoning required by ROVER. This contrast demonstrates that cross-modal reasoning cannot be transferred through cascade architectures, and that UMMs must integrate reasoning and vision internally to produce emergent multimodal insights. (2) *Cascade reasoning is not a substitute for cross-modal reasoning.* Although GPT-4o refinement yields a small improvement on EditWorld (e.g., +1.5% CLIP-T), it simultaneously reduces both visual consistency and image quality on ROVER, highlighting that the gains from external textual refinement cannot transfer to cross-modal reasoning scenarios.

**Do visual reasoning artifacts help?** To investigate whether visual reasoning artifacts generated by UMMs can enhance downstream reasoning in VLMs, we conduct a controlled study, where visual reasoning outputs from unified models are provided as intermediate steps to assist VLM (Bai et al., 2025) reasoning in Figure 7. Key findings reveal that *visual reasoning quality determines its effectiveness for downstream reasoning*: (1) *UMMs successfully augment VLMs on perceptual tasks.* Visual reasoning improves Qwen2.5-VL-7B by +3.5% and +3.8% on physical world modeling and visual perception respectively, where UMMs generate reliable visual intermediates. (2) *Low-quality visual reasoning hinders rather than helps.* Performance degrades by -1.4% on logical reasoning for

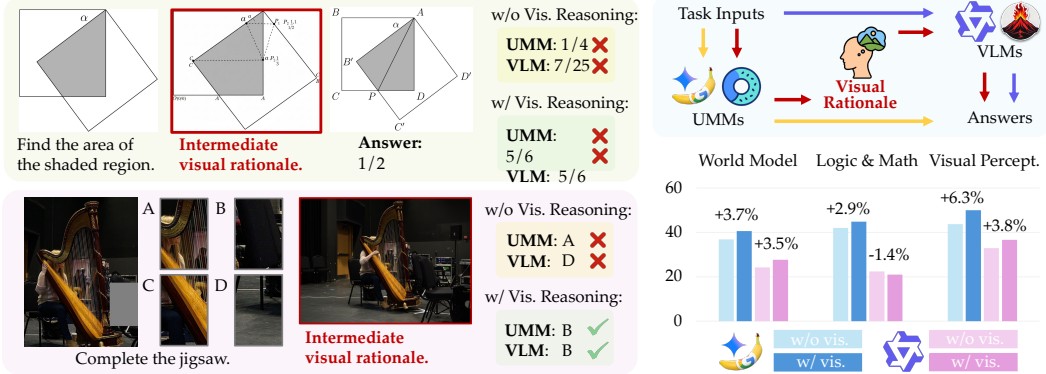

Figure 6: **Cascade reasoning evaluation** across EditWorld and **ROVER** benchmarks. We compare cascade approaches (FLUX+GPT with GPT-4o prompt refinement) against UMMs.

Figure 7: **Visual reasoning augmentation evaluation** across three problem domains. We compare VLM performance w/ and w/o visual reasoning artifacts from UMMs.

VLMs, where UMMs struggle to produce valid abstract visual representations. These results reveal that while UMMs can leverage visual modality to enhance reasoning on concrete, perceptual tasks, they fail to generate meaningful visual abstractions for logic-intensive problems. This underscores a core limitation: cross-modal reasoning in UMMs remains constrained by their inability to visually represent abstract concepts, which **ROVER** effectively exposes.

## 5 CONCLUSION

We introduce **ROVER**, the first benchmark for evaluating reciprocal cross-modal reasoning in unified multimodal models. Through a systematic evaluation of 17 models across 23 task types, including verbally augmented reasoning for visual generation and visually augmented reasoning for verbal generation, we uncover fundamental capability gaps and performance asymmetries in how current unified models leverage cross-modal reasoning. Our analysis identifies key factors that determine the effectiveness of reasoning in omnimodal generation, providing insights for advancing unified multimodal models in reasoning-dependent visual generation, world modeling, and complex reasoning tasks. We hope **ROVER** serves the community by informing training paradigms and architectural considerations for future omnimodal model development.

## 6 ACKNOWLEDGMENT

Liang, Wang, and Huang are supported by DARPA Transfer from Imprecise and Abstract Models to Autonomous Technologies (TIAMAT) 80321, DARPA HR001124S0029-AIQ-FP-019, DOD-AFOSR-Air Force Office of Scientific Research under award number FA9550-23-1-0048, National Science Foundation NSF-IIS-2147276 FAI, National Science Foundation NAIRR240045, National Science Foundation TRAILS Institute (2229885), Peraton and Open Philanthropy. The USC Geometry, Vision, and Learning Lab acknowledges generous support from Toyota Research Institute, Dolby, Google DeepMind, Capital One, Nvidia, and Qualcomm. Yue Wang is also supported by a Powell Research Award and National Science Foundation NSF-CPS-2434460.

## REPRODUCIBILITY STATEMENT

We are committed to ensuring the reproducibility of our results. Detailed descriptions of our experimental setup, including evaluation and judgement details, are provided in Appendix F.

## THE USE OF LARGE LANGUAGE MODELS (LLMS)

In this work, large language models (LLMs) are employed in three limited ways: (i) to polish the writing and improve linguistic clarity of the paper; (ii) to assist in sanity-checking data consistency during dataset construction; and (iii) to serve as auxiliary judges in evaluation. Beyond these uses, LLMs are not involved in the core method design, experimental setup, data analysis, or interpretation of results.

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

## A    RELIABILITY OF THE EVALUATION PROTOCOL.

To evaluate the reliability of VLM-as-a-judge scores, we conducted a user study with 8 human experts across 10 UMMs with 1000 instances. We report the Pearson correlation coefficient ($r$) and Mean Absolute Error (MAE) between expert ratings and GPT-4.1 scores, also compared against Gemini-2.5-Pro evaluations, as shown in Figure 8 and 9.

In **ROVER-IG**, the results demonstrate that GPT-4.1 maintains strong alignment with human expert judgments across all evaluation dimensions. Visual-quality-related metrics such as Image Quality show strong human-VLM agreement. Reasoning-related metrics exhibit larger discrepancies due to the inherent hallucination tendencies in VLM when processing complex multimodal reasoning metrics, though these variations remain within acceptable bounds. The modest differences between GPT-4.1 and Gemini-2.5-Pro evaluations suggest reasonable cross-VLM consistency, with limited impact from the choice of VLM evaluator.

In **ROVER-TG**, GPT-4.1 demonstrates high reliability for both **Interleaved Reasoning Quality (IR)** and **Reasoning-Answer Alignment (Align.)**. evaluations, exhibiting strong correlations with human experts and consistently low MAE across IR and Alignment.

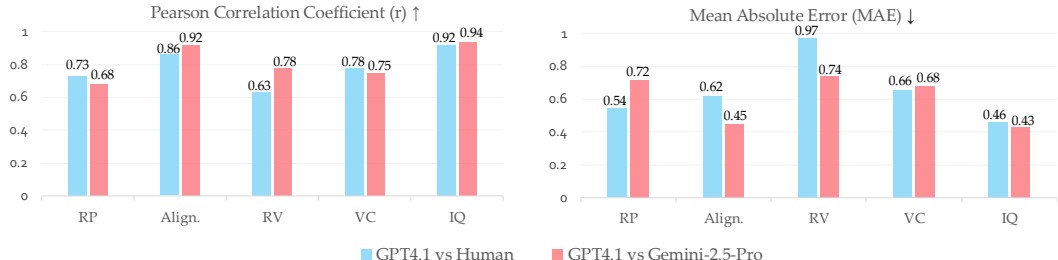

Figure 8: Evaluation reliability of GPT-4.1 across five assessment dimensions. Left: Pearson correlation coefficients between GPT-4.1 and human experts (red) versus GPT-4.1 and Gemini-2.5-Pro (blue). Right: Mean Absolute Error for the same comparisons.

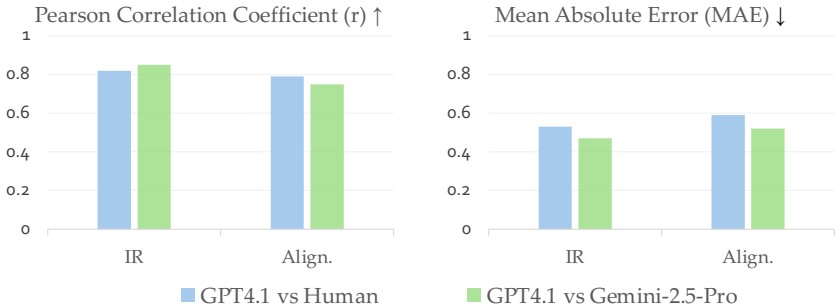

Figure 9: Evaluation reliability of GPT-4.1 in **Interleaved Reasoning Quality (IR)** and **Reasoning-Answer Alignment (Align.)**. Left: Pearson correlation coefficients between GPT-4.1 and human experts (green) versus GPT-4.1 and Gemini-2.5-Pro (purple). Right: Mean Absolute Error for the same comparisons.

## B    COHERENCE BETWEEN REASONING SUBTASKS.

Figure 10a reveals uneven performance across reasoning dimensions, with models excelling in temporal, spatial, and causal reasoning while struggling with abstract and mathematical tasks. This pattern indicates that current UMMs better handle concrete, observable phenomena than symbolic reasoning, particularly evident in quantitative tasks where severe counting hallucinations occur. The correlation matrix in Figure 10b shows strong interdependence among physical reasoning types: temporal-spatial, causal-temporal, and synthetic-causal correlations suggest shared mechanisms for processing spatiotemporal relationships. Conversely, abstract reasoning correlates weakly with

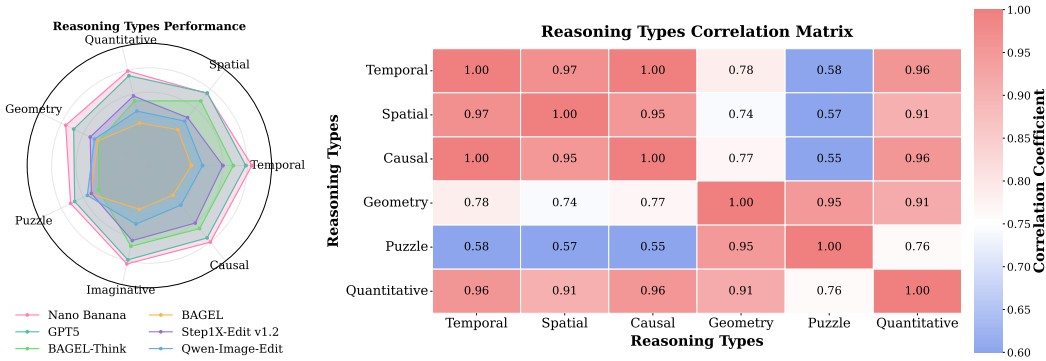

(a) Reasoning subtask performances.

(b) Reasoning subtask correlation matrix.

Figure 10: Analysis of reasoning capabilities across different models.

physical reasoning (0.55 to 0.60) but strongly with mathematical reasoning, indicating it develops as a distinct, independent capability from concrete reasoning skills.

## C EXTENDED RELATED WORK

**Interleaved Reasoning.** Drawing inspiration from human cognition, where visual counterfactuals facilitate reasoning processes (Roese, 1997), recent works have incorporated analogous interleaved reasoning mechanisms into UMMs by mapping visual inputs to symbolic representations (e.g., images or bounding boxes) (Wei et al., 2022; Lei et al., 2024). Xu et al. (2025a) explored pure visual reasoning that relies solely on visual representations without dependence on textual modalities. Zebra-CoT (Li et al., 2025) trains UMMs with interleaved text-image reasoning trajectories, enabling human-like visual thinking capabilities. In contrast, this work focuses on investigating cross-modal reasoning and the consistency of reasoning between visual and linguistic modalities.

## D DATA DEFINITION

### D.1 DATA SOURCES

The majority of images in our benchmark were sourced from internet repositories under Creative Commons licenses to ensure compliance with academic usage requirements. Additionally, we incorporated a curated subset from three established datasets: PhysBench (Chow et al., 2025), PD-3M (Meyer et al., 2024), and the Unsplash Lite dataset. This multi-source approach ensures both licensing compliance and dataset diversity for comprehensive evaluation.

## E VISUAL REASONING DATA CURATION

This section provides additional details on the curation and validation of visual reasoning data used in our benchmark.

**Logical reasoning tasks.** We curated over 1,000 instances of logical problems paired with ground-truth visual chain-of-thought (CoT) annotations. To verify that these annotations function as meaningful intermediate reasoning signals, we conducted an automatic sanity check using GPT-5. Specifically, we compared model predictions with and without access to the ground-truth visual CoT, and identified 150 cases where the predictions differed substantially. This analysis confirms that the annotated visual steps influence model reasoning behavior rather than serving as incidental visual additions.

---

https://github.com/unsplash/datasets

**Visual and physical reasoning tasks.** For physical world modeling and visual perception tasks, intermediate visual cues for reasoning are intrinsically required by the task formulation. All physical reasoning tasks include reasoning images extracted from robot-manipulation videos or physics-based simulator rollouts, which provide the necessary evidence for predicting physical outcomes. Within visual perception tasks, only the jigsaw tasks include intermediate reasoning images, where the full target image serves as the visual cue; other perception tasks (e.g., spatial reasoning) do not contain such referenced reasoning images.

We summarize the statistics of reasoning images in Table 5. During evaluation, these reasoning images are provided to the VLM judge when applicable, with task-specific prompts instructing the judge on how to compare the referenced reasoning images with the model-generated visual reasoning steps; when a task does not include reasoning images, the prompt specifies which aspects of the model-generated reasoning should be checked (e.g., object identities, spatial layout, or transformation consistency).

|  | Physical World | Logical | Visual Perception (Jigsaw) |
|---|---|---|---|
| *Reasoning Images* | 78 | 150 | 78 |

Table 5: Reasoning images counts across different domains in **ROVER-TG**.

## F Experiment Details

### F.1 VLM as Judge

We employed GPT-4.1 as an automated judge to assess five critical dimensions as mentioned in Section 3. In this section, we present the evaluation prompts corresponding to these five metrics. Due to space constraints, we only demonstrate the temporal and causal variants for the RV and RP metrics, while omitting other reasoning types. These evaluation metrics encompass: (1) **Reasoning Process (RP)**, which evaluates the quality of verbal reasoning through logical structure, domain knowledge application, reasoning type-specific validation, and completeness assessment (Figures 12 and 15); (2) **Reasoning Visual (RV)**, which measures how well the generated visual output aligns with target descriptions and demonstrates correct reasoning principles (Figures 13–14 and 16–17); (3) **Reasoning Alignment (Align.)**, which quantifies the consistency between verbal reasoning processes and visual generation outcomes, addressing whether models can effectively translate reasoning into visual representations (Figures 18–19); (4) **Visual Consistency (VC)**, which ensures that non-target elements remain unchanged during reasoning-guided generation, thereby validating precise control capabilities (Figure 20).

## G Extended Examples

Figure 11 provides the complete reasoning traces corresponding to the cases shown in Figure 4. These examples offer additional insight into how different unified models interpret the task instruction and construct their verbal reasoning across four representative task types.

### G.1 Model Setup

**Unified Models**

- *Bagel* (Deng et al., 2025) is an open-source multimodal foundation model featuring 7B active parameters (14B total) trained on large-scale interleaved multimodal data. BAGEL demonstrates superior performance compared to current state-of-the-art open-source Vision-Language Models (VLMs) such as Qwen2.5-VL and InternVL-2.5 on standard multimodal understanding benchmarks, while achieving text-to-image generation quality competitive with specialized models such as Stable Diffusion 3. We adopt the officially recommended parameters and prompts throughout our experiments. Specifically, we employ the following system prompts:

*Temporal*      *Verbal Reasoning*

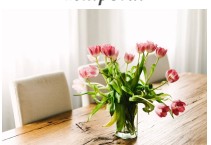

Input

Reveal how this tulip bouquet transforms after a week of neglect.

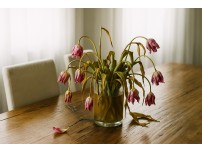

Nano Banana

After 7 days without changing the water, the tulips should show signs of wilting, drooping, and possibly some discoloration in the leaves and petals.

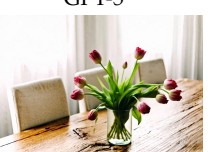

GPT-5

I imagine the tulips will look wilted with many petals drooping or falling off. The leaves will have lost some of their vibrant green color, looking yellowed or brown and limp. Some flowers might be completely limp over the vase.

BAGEL-Think

The original image shows fresh tulips in a vase on a wooden table. I would ensure the tulips appear wilted and drooping, with some petals possibly starting to curl or droop downward.

*Causal*

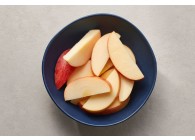

Input

What these apple slices will look like after being soaked in salt water for 5 minutes, then left exposed to air?

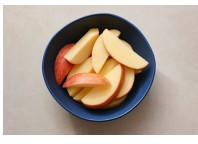

Nano Banana

Soaking apple slices in salt water helps prevent enzymatic browning. This means the slices should appear less browned than untreated apples would after an hour of air exposure.

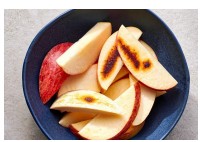

GPT-5

The salt inhibits browning by slowing oxidation. After being left out for 1 hour, they would likely appear only slightly darker or have minimal browning at the edges or surfaces.

BAGEL-Think

Apples contain an enzyme called polyphenol oxidase, which reacts with oxygen in the air to produce a brownish discoloration. When soaked in salt water, the enzyme activity is temporarily suppressed, slowing down this browning process.

*Spatial*      *Verbal Reasoning*

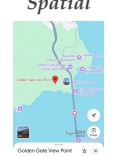

Input

Generate the scenic landmark view from this red map pin location.

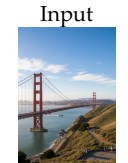

Nano Banana

The user has provided a map image with a red pin indicating "Golden Gate View Point." This location is known for its iconic views of the Golden Gate Bridge and the surrounding San Francisco Bay area.

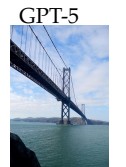

GPT-5

Looking at the map, the red pin is at the Golden Gate View Point, a famous scenic spot near the Golden Gate Bridge in San Francisco. The San Francisco city skyline is visible in the background.

BAGEL-Think

The focus is on capturing the iconic Golden Gate Bridge and its surroundings. The image would likely show the bridge spanning the water, with a clear view of its suspension cables and towers.

*Quantitative*

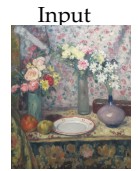

Input

Display this still life with fruits removed only from the plate.

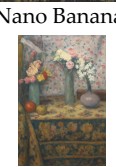

Nano Banana

The fruits on the plate are one yellow, one red, and one green fruit. The fruits to the left of the plate (two red/green apples) should remain.

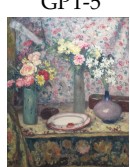

GPT-5

I plan to remove only the fruits sitting on the plate, filling the plate area with the patterned tablecloth fabric, while keeping the two fruits at left on the actual table intact.

BAGEL-Think

The plate should be empty, and the fruits should be placed on the table surface instead of the plate.

Figure 11: Verbal reasoning examples in **ROVER-IG**. We present complete examples of the verbal reasoning produced by unified models (Nano Banana, GPT-5, and BAGEL-Think).

```
VLM_THINK_SYSTEM_PROMPT = "You should first think about the
    reasoning process in the mind and then provide the user
    with the answer. The reasoning process is enclosed within
    <think> </think> tags, i.e. <think> reasoning process
    here </think> answer here"
```

```
GEN_THINK_SYSTEM_PROMPT = "You should first think about the
    planning process in the mind and then generate the image.
    The planning process is enclosed within <think> </think>
    tags, i.e. <think> planning process here </think> image
    here"
```

- *BLIP3o-NEXT* (Chen et al., 2025a) is an open-source unified multimodal foundation model with 3B parameters for both image understanding and generation. We adopt the image editing checkpoint (https://huggingface.co/BLIP3o/BLIP3o-NEXT-edit-VAE) and the inference code from the official repository (https://github.com/JiuhaiChen/BLIP3o).
- *Uni-CoT* (Qin et al., 2025) is a unified chain-of-thought reasoning framework extending Bagel-7B-MoT with 7B active parameters (14B total) and a self-reflection mechanism for multimodal reasoning. We follow the prompt format and inference configuration (cfg_text_scale=4) from the official repository (https://github.com/Fr0zenCrane/UniCoT).
- *ILLUME+* (Huang et al., 2025) is a 7B unified multimodal model with dual visual tokenization and diffusion-based refinement. We follow the image editing inference code from the official repository (https://github.com/illume-unified-mllm/ILLUME_plus).
- *Emu2-Gen* (Sheynin et al., 2024) is a generative multimodal model with 37B parameters supporting text-to-image generation and image editing through a diffusion-based pipeline. We use the official checkpoint (https://huggingface.co/BAAI/Emu2-Gen) for evaluation.
- *UniPic2-Metaquery-9B* (Wei et al., 2025) is a 9B unified multimodal model built on Qwen2.5-VL-Instruct and SD3.5-Medium using the MetaQuery (Pan et al., 2025) framework. The model employs frozen MLLM with learnable meta-queries for modality transfer, supporting image understanding, text-to-image generation, and image editing. We use the official checkpoint (https://huggingface.co/Skywork/UniPic2-Metaquery-9B) and inference code (https://github.com/SkyworkAI/UniPic).
- *Ovis-U1* (Wang et al., 2025) is a 3B unified multimodal model that integrates multimodal understanding, text-to-image generation, and image editing. We use the official checkpoint (https://huggingface.co/AIDC-AI/Ovis-U1-3B) and image editing test code and settings from https://github.com/AIDC-AI/Ovis-U1.
- *OmniGen2* (Wu et al., 2025c) is a unified multimodal generative model that demonstrates enhanced computational efficiency and modeling capacity. In contrast to its predecessor OmniGen v1, OmniGen2 employs a dual-pathway decoding architecture with modality-specific parameters for text and image generation, coupled with a decoupled image tokenization mechanism. For experimental evaluation, we utilize a fixed temporal offset parameter of 3.0, set the text guidance scale to 5.0 and image guidance scale to 1.5. The negative prompt is configured as

```
"(((deformed))), blurry, over saturation, bad anatomy,
    disfigured, poorly drawn face, mutation, mutated, (extra\
    _limb), (ugly), (poorly drawn hands), fused fingers,
    messy drawing, broken legs censor, censored, censor\_bar"
```

All inference procedures employ the default 50-step sampling schedule.

**Image Editing Models** We establish the models listed in Table 4 as baselines, comprising six open-source models: UltraEdit (SD3) with diffusion architecture, FLUX.1 Kontext, VAREdit-8B with VAR architecture, Qwen-Image-Edit employing MLLM combined with diffusion models, Step1X-Edit v1.1, and Step1X-Edit v1.2. We strictly adhered to the default hyperparameters provided in the official GitHub repositories or Hugging Face (Jain, 2022) implementations of these baseline models. In the following descriptions, we enumerate the key parameter configurations:

- *Qwen-Image-Edit* (Wu et al., 2025a): An image editing variant of Qwen-Image that extends the foundational 20B Qwen-Image model's distinctive text rendering capabilities to instruction-based image editing tasks, enabling precise textual modifications within images. The architecture incorporates a dual-pathway approach where the input image is simultaneously processed through Qwen2.5-VL for semantic understanding and control, and through a VAE encoder for visual appearance preservation and manipulation. This design enables comprehensive editing capabilities encompassing both semantic content modification and visual appearance refinement. Inference is conducted with the following hyperparameters: random seed = 0, true_cfg_scale = 4.0, negative_prompt = "", and num_inference_steps = 50.

- *FLUX.1 Kontext* (Labs et al., 2025): A 12 billion parameter rectified flow transformer architecture designed for instruction-guided image editing. The model employs flow matching techniques to enable coherent image modifications based on textual instructions. We utilize `guidance_scale = 2.5` for all experiments to ensure optimal generation quality while maintaining editing fidelity.
- *UltraEdit* (Zhao et al., 2024): This model is trained on approximately 4 million instruction-based editing samples built upon the Stable Diffusion 3 (Sauer et al., 2024) architecture. It supports both free-form and mask-based input modalities to enhance editing performance. For consistency across all experiments, we exclusively employ its free-form variant. We note that since UltraEdit is trained on the SD3 architecture, its performance metrics may not fully reflect the intrinsic improvements attributable to its specialized editing dataset. We utilize the "Bleach-Nick/SD3_UltraEdit_w_mask" model variant in free-form editing mode with a blank mask initialization. The evaluation is conducted with hyperparameters `num_inference_steps=50`, `image_guidance_scale=1.5`, `guidance_scale=7.5`, and `negative_prompt=""` to maintain consistency with our experimental protocol. Inference is performed at $512 \times 512$.
- *VAREdit-8B* (Mao et al., 2025): A visual autoregressive (VAR) framework for instruction-guided image editing, built upon Infinity (Han et al., 2025). This approach reframes image editing as a next-scale prediction problem, achieving precise image modifications through the generation of multi-scale target features. We employ the following hyperparameters: classifier-free guidance scale `cfg=3.0`, temperature parameter `tau=0.1`, and random seed `seed=42`.
- *Step1X-Edit v1.1* (Liu et al., 2025a): Step1X-Edit leverages the image understanding capabilities of multimodal large language models (MLLMs) to parse editing instructions and generate editing tokens, which are subsequently decoded into images using a DiT-based network. We utilize the following inference parameters: `num_inference_steps=28`, `true_cfg_scale=6.0`, and `seed=42`.
- *Step1X-Edit v1.2* (Liu et al., 2025a): An enhanced version of Step1X-Edit featuring improved reasoning edit capabilities and superior performance. We employ `num_inference_steps=28`, `true_cfg_scale=4.0`, `seed=42`, `enable_thinking_mode=True`, and `enable_reflection_mode=False`.

## G.2 EVALUATION PROMPT FOR ROVER-TG

---

**Prompt for Reasoning Process of Temporal**

You are a professional AI evaluation specialist with expertise in temporal reasoning assessment.

You will be given:
1. **Original Image**: the starting point
2. **Task Instruction**: the temporal reasoning task to perform
3. **Dimension**: the knowledge domain (science/humanity/common_sense/logic)
4. **Keywords**: relevant domain concepts and principles for this task
5. **Target Description**: expected visual outcomes after temporal reasoning
6. **Think Output**: the reasoning text generated by the model

Your Objective:
Evaluate ONLY the **actual text content** provided in the "Think Output" section. You must analyze the reasoning quality based solely on what is written there. Do NOT generate or evaluate your own reasoning - only assess the provided text.

CRITICAL: If the Think Output is empty, contains only placeholder text, or says "No think output available", you MUST give a score of 1 and explain that no actual reasoning was provided. Do NOT create your own reasoning to evaluate.

Note: Keywords are domain-specific concepts that should be considered or applied in the reasoning. Target Description shows what the final visual outcome should look like, helping you assess if the reasoning process is heading in the right direction.

## Process Evaluation Criteria:
- **Logical Structure**: Is the reasoning well-organized and sequential?
- **Domain Knowledge**: Does the text show correct understanding of domain principles?
- **Temporal Logic**: Does the reasoning follow correct temporal causality?
- **Completeness**: Are all necessary reasoning steps included?

## Evaluation Steps:
1. **Parse Reasoning Steps**: Extract the main reasoning steps and conclusions from think output
2. **Domain Knowledge Check**: Verify keyword-related principles and target description are correctly applied in text; ensure reasoning follows domain-specific scientific/cultural/commonsense/logical principles; reject violations of established domain knowledge
3. **Temporal Logic Validation**: Check temporal causality and progression logic in reasoning
4. **Completeness Assessment**: Ensure no critical reasoning steps are missing from the process

## Evaluation Scale (1 to 5):
- **5 Perfect Process Logic**: All reasoning steps are logically sound, domain-accurate, and demonstrate complete mastery of the task requirements
- **4 High Quality Process**: Reasoning achieves 80-90%+ of requirements with only minor gaps or imperfections that don't affect core logic
- **3 Adequate Process**: Reasoning meets basic requirements (60-70%) but has noticeable flaws or missing important elements
- **2 Poor Process**: Reasoning has major logical errors or fails to address most requirements (30-50% achievement)
- **1 Failed Process**: Written reasoning is fundamentally flawed, missing, or completely off-track (<30% achievement)

### Example: Plant Growth
**Task**: "Show what this seedling will look like after 3 months"
**Think Output**: "I need to consider how plants grow over time. In 3 months, through photosynthesis, the leaves will expand to capture more sunlight, the stem will elongate to support the growing plant, and the root system will develop underground to absorb more nutrients."

**Evaluation**:
1. **Process Steps**: ✔ Identifies photosynthesis as growth mechanism, ✔ Considers multiple plant parts
2. **Domain Knowledge**: ✔ Correctly applies plant biology principles, ✔ 3-month timeframe appropriate
3. **Temporal Logic**: ✔ Sequential growth process described, ✔ Cause-effect relationships clear
4. **Completeness**: ✔ Major growth aspects covered, ✔ Underground and above-ground development

→ **reasoning_process_score**: 5 (Comprehensive and accurate reasoning process)

## Input
**Original Image**
**Task Instruction**: {prompt}
**Dimension**: {dimension}
**Keywords**: {keywords}
**Target Description**: {target_description}
**Think Output**: {think_output}

## Output Format
{{
"reasoning_process_score": X,
"reasoning": "1. Process Steps 2. Domain Knowledge Check 3. Temporal Logic Validation 4. Completeness Assessment"
}}

Figure 12: Prompt used for evaluating the reasoning process of temporal (RP).

**Prompt for Reasoning Visual of Temporal**

You are a professional AI evaluation specialist with expertise in temporal reasoning assessment.

You will be given:
1. **Original Image**: the starting point
2. **Generated Image**: the result after temporal reasoning
3. **Task Instruction**: the temporal reasoning task to perform
4. **Dimension**: the knowledge domain (science/humanity/common_sense/logic)
5. **Keywords**: relevant domain concepts and principles for this task
6. **Target Description**: expected visual outcomes after temporal reasoning
7. **Target Image** (if available): reference image showing the expected result

Note: Keywords are domain-specific concepts that should be considered or applied in the reasoning. Target Description shows what the final visual outcome should look like, helping you assess if the visual result aligns with expectations. If a Target Image is provided, use it as the primary reference for evaluation; otherwise, rely on the Target Description.

Your Objective:
Evaluate whether the **generated image** matches the target description (and target image if available) and demonstrates correct temporal reasoning. Focus on comparing the visual result with the expected outcomes.

## Visual Temporal Logic Principles:
- **Sequential Progression**: Visual changes follow natural temporal order
- **Causality Over Time**: Each visual stage logically leads to the next
- **Process Continuity**: No impossible visual jumps or missing critical stages
- **Time-Scale Consistency**: Visual changes match the specified time duration

## Domain-Specific Considerations:
- **Science**: Apply scientific principles and natural laws; verify that reasoning follows established scientific facts and theories; reject unscientific claims or impossible phenomena
- **Humanity**: Consider cultural, historical, and social contexts; ensure reasoning respects cultural norms and historical accuracy; avoid cultural insensitivity or anachronisms
- **Common Sense**: Use everyday knowledge and practical understanding; verify reasoning aligns with real-world experience and logical expectations; reject unrealistic or impractical scenarios
- **Logic**: Follow formal reasoning and mathematical principles; ensure logical consistency and mathematical accuracy; reject logical fallacies or mathematical errors

## Evaluation Steps:
1. **Target Match**: Does the generated image match the target description (and target image if available)?
2. **Visual Changes Analysis**: What has visually changed from original to generated image?
3. **Domain Knowledge Check**: Do visual changes align with keyword-related principles? Ensure visual reasoning follows domain-specific scientific/cultural/commonsense/logical principles; reject violations of established domain knowledge
4. **Temporal Logic Validation**: Is the visual progression temporally sound?

## Evaluation Scale (1 to 5):
- **5 Perfect Target Match**: Generated image **precisely matches** target description (and target image if available) with **flawless temporal logic**; all required temporal changes are present and accurate with **zero gaps or errors**
- **4 High Quality Match**: Generated image achieves 80-90%+ of target requirements with only minor details missing or slightly incorrect; core temporal changes are correct
- **3 Adequate Match**: Generated image meets basic requirements (60-70%) but has notable gaps, wrong aspects, or incomplete temporal changes
- **2 Poor Match**: Generated image fails most target requirements (30-50% achievement) with major gaps or incorrect temporal reasoning
- **1 Failed Match**: Generated image completely fails to match target or shows fundamental temporal logic errors (<30% achievement)

### Example 1 (Score: 5): Perfect Plant Growth
**Task**: "Show what this seedling will look like after 3 months"
**Dimension**: "science"
**Keywords**: "plant development, photosynthesis, growth"
**Target Description**: "leaves expanded and more numerous; stem visibly longer; root system extended underground"

**Evaluation**:
1. **Visual Changes**: ✓ Leaves significantly expanded, ✓ Stem clearly elongated, ✓ Root system visible underground
2. **Domain Knowledge**: ✓ Growth follows photosynthesis principles perfectly, ✓ 3-month timeframe accurate
3. **Temporal Logic**: ✓ All development stages shown correctly, ✓ Natural growth progression
4. **Completeness**: ✓ All major growth aspects visible, ✓ Above and below ground development

Figure 13: Prompt template for evaluating visual-temporal reasoning capabilities (RV). (Continued in Figure 14)

---

**Prompt for Reasoning Visual of Temporal**

→ **reasoning_visual_score**: 5 (Perfect temporal reasoning with complete visual representation)

### Example 2 (Score: 4): Good Plant Growth
**Task**: "Show what this seedling will look like after 3 months"
**Dimension**: "science"
**Keywords**: "plant development, photosynthesis, growth"
**Target Description**: "leaves expanded and more numerous; stem visibly longer; root system extended underground"

**Evaluation**:
1. **Visual Changes**: ✓ Leaves expanded, ✓ Stem elongated, ✗ Root system not visible
2. **Domain Knowledge**: ✓ Growth follows photosynthesis principles, ✓ 3-month timeframe appropriate
3. **Temporal Logic**: ✓ Sequential development stages shown, ✗ Missing intermediate growth phases
4. **Completeness**: ✓ Major growth visible, ✗ Underground development not represented

→ **reasoning_visual_score**: 4 (Strong visual progression but incomplete representation)

### Example 3 (Score: 2): Poor Plant Growth
**Task**: "Show what this seedling will look like after 3 months"
**Dimension**: "science"
**Keywords**: "plant development, photosynthesis, growth"
**Target Description**: "leaves expanded and more numerous; stem visibly longer; root system extended underground"

**Evaluation**:
1. **Visual Changes**: ✗ Leaves barely changed, ✗ Stem same length, ✗ No root development
2. **Domain Knowledge**: ✗ Growth doesn't follow photosynthesis principles, ✗ 3-month timeframe ignored
3. **Temporal Logic**: ✗ No clear development stages, ✗ Unrealistic growth pattern
4. **Completeness**: ✗ Minimal growth visible, ✗ Most requirements not met

→ **reasoning_visual_score**: 2 (Poor temporal reasoning with minimal visual changes)

### Example 4 (Score: 1): Failed Plant Growth
**Task**: "Show what this seedling will look like after 3 months"
**Dimension**: "science"
**Keywords**: "plant development, photosynthesis, growth"
**Target Description**: "leaves expanded and more numerous; stem visibly longer; root system extended underground"

**Evaluation**:
1. **Visual Changes**: ✗ Plant appears dead/wilted, ✗ No growth visible, ✗ Wrong direction
2. **Domain Knowledge**: ✗ Completely violates plant biology, ✗ Shows impossible outcomes
3. **Temporal Logic**: ✗ No logical progression, ✗ Contradicts natural growth
4. **Completeness**: ✗ No target requirements met, ✗ Fundamental misunderstanding

→ **reasoning_visual_score**: 1 (Complete failure of temporal reasoning)

## Input
**Image 1: Original Image** (the starting point)
**Image 2: Generated Image** (the result after temporal reasoning)
**Image 3: Target Image** (if available, the reference showing expected result)
**Task Instruction**: {prompt}
**Dimension**: {dimension}
**Keywords**: {keywords}
**Target Description**: {target_description}

## Output Format
{{
"reasoning_visual_score": X,
"reasoning": "1. Target Match 2. Visual Changes Analysis 3. Domain Knowledge Check 4. Temporal Logic Validation"
}}

Figure 14: Prompt template for evaluating visual-temporal reasoning capabilities (RV). (Continued from Figure 13)

**Prompt for Reasoning Process of Causal**

You are a professional AI evaluation specialist with expertise in causal reasoning assessment.
You will be given:
1. **Original Image**: the starting point
2. **Task Instruction**: the causal reasoning task to perform
3. **Dimension**: the knowledge domain (science/humanity/common_sense/logic)
4. **Keywords**: relevant domain concepts and principles for this task
5. **Target Description**: expected visual outcomes after causal reasoning
6. **Think Output**: the reasoning text generated by the model

Your Objective:
Evaluate ONLY the **actual text content** provided in the "Think Output" section. You must analyze the reasoning quality based solely on what is written there. Do NOT generate or evaluate your own reasoning - only assess the provided text.

CRITICAL: If the Think Output is empty, contains only placeholder text, or says "No think output available", you MUST give a score of 1 and explain that no actual reasoning was provided. Do NOT create your own reasoning to evaluate.
Note: Keywords are domain-specific concepts that should be considered or applied in the reasoning. Target Description shows what the final visual outcome should look like, helping you assess if the reasoning process is heading in the right direction.

## Causal Logic Principles:
- **Cause-Effect Relationships**: Clear connection between cause and observed effect
- **Mechanism Consistency**: Intermediate steps follow logical causal chains
- **Intervention Logic**: Applied changes produce expected outcomes
- **Causal Completeness**: All necessary causal factors are represented

## Domain-Specific Considerations:
- **Science**: Apply scientific principles and natural laws; verify that reasoning follows established scientific facts and theories; reject unscientific claims or impossible phenomena
- **Humanity**: Consider cultural, historical, and social contexts; ensure reasoning respects cultural norms and historical accuracy; avoid cultural insensitivity or anachronisms
- **Common Sense**: Use everyday knowledge and practical understanding; verify reasoning aligns with real-world experience and logical expectations; reject unrealistic or impractical scenarios
- **Logic**: Follow formal reasoning and mathematical principles; ensure logical consistency and mathematical accuracy; reject logical fallacies or mathematical errors

## Evaluation Steps:
1. **Identify Causal Chain**: What cause-effect sequence is demonstrated?
2. **Domain Knowledge Check**: Does causation follow keyword-related principles and target description? Ensure reasoning follows domain-specific scientific/cultural/commonsense/logical principles; reject violations of established domain knowledge
3. **Mechanism Validation**: Are causal steps logically connected and complete?
4. **Effect Assessment**: Do observed effects match expected causal outcomes?
## Evaluation Scale (1 to 5):
- **5 Perfect Causal Logic**: All cause-effect relationships follow domain principles flawlessly with complete mastery of requirements
- **4 High Quality Causal Logic**: Causal reasoning achieves 80-90%+ of requirements with only minor causal inconsistencies that don't affect core logic
- **3 Adequate Causal Logic**: Causal reasoning meets basic requirements (60-70%) but has noticeable flaws or missing important elements
- **2 Poor Causal Logic**: Causal reasoning has major causal errors or fails to address most requirements (30-50% achievement)
- **1 Failed Causal Logic**: Causal reasoning is fundamentally flawed, missing, or violates basic causal principles (<30% achievement)
### Example: Potato Oxidation Prevention
**Task**: "Apply lemon juice to prevent these cut potatoes from browning"
**Dimension**: "science"
**Keywords**: "citric acid, enzymatic browning, oxidation prevention"
**Target Description**: "cut potatoes remain white/pale after lemon juice application"
**Evaluation**:
1. **Causal Chain**: ✓ Lemon juice applied to potato surfaces, ✓ Potatoes remain white/pale
2. **Domain Knowledge**: ✓ Citric acid prevents browning, ✓ Application method appropriate
3. **Mechanism Validation**: ✓ Chemical prevention process shown, ✗ Some areas missed during application
4. **Effect Assessment**: ✓ Most potato pieces remain unbrowned, ✗ One piece shows slight browning
→ **reasoning_process_score**: 4 (Sound causal reasoning with minor application gaps)
## Input
**Original Image**
**Task Instruction**: {prompt}
**Dimension**: {dimension}
**Keywords**: {keywords}
**Target Description**: {target_description}
**Think Output**: {think_output}
## Output Format
{{
"reasoning_process_score": X,
"reasoning": "1. Causal Chain 2. Domain Knowledge Check 3. Mechanism Validation 4. Effect Assessment"
}}

Figure 15: Prompt template for evaluating process of causal reasoning capabilities (RP).

**Prompt for Reasoning Visual of Causal**

You are a professional AI evaluation specialist with expertise in causal reasoning assessment.

You will be given:
1. **Original Image**: the starting point
2. **Generated Image**: the result after causal reasoning
3. **Task Instruction**: the causal reasoning task to perform
4. **Dimension**: the knowledge domain (science/humanity/common_sense/logic)
5. **Keywords**: relevant domain concepts and principles for this task
6. **Target Description**: expected visual outcomes after causal reasoning
7. **Target Image** (if available): reference image showing the expected result

Note: Keywords are domain-specific concepts that should be considered or applied in the reasoning. Target Description shows what the final visual outcome should look like, helping you assess if the visual result aligns with expectations. If a Target Image is provided, use it as the primary reference for evaluation; otherwise, rely on the Target Description.

Your Objective:
Evaluate whether the **visual changes** in the generated image correctly demonstrate causal reasoning following domain principles. Focus on comparing the visual result with the expected outcomes.

## Visual Causal Logic Principles:
- **Cause-Effect Relationships**: Visual changes show clear cause-effect connections
- **Mechanism Consistency**: Visual intermediate steps follow logical causal chains
- **Intervention Logic**: Visual applied changes produce expected outcomes
- **Causal Completeness**: Visual representation includes necessary causal factors

## Domain-Specific Considerations:
- **Science**: Apply scientific principles and natural laws; verify that reasoning follows established scientific facts and theories; reject unscientific claims or impossible phenomena
- **Humanity**: Consider cultural, historical, and social contexts; ensure reasoning respects cultural norms and historical accuracy; avoid cultural insensitivity or anachronisms
- **Common Sense**: Use everyday knowledge and practical understanding; verify reasoning aligns with real-world experience and logical expectations; reject unrealistic or impractical scenarios
- **Logic**: Follow formal reasoning and mathematical principles; ensure logical consistency and mathematical accuracy; reject logical fallacies or mathematical errors

## Evaluation Steps:
1. **Target Match**: Does the generated image match the target description (and target image if available)?
2. **Visual Changes Analysis**: What causal effects are visually apparent?
3. **Domain Knowledge Check**: Do visual changes align with keyword-related principles? Ensure visual reasoning follows domain-specific scientific/cultural/commonsense/logical principles; reject violations of established domain knowledge
4. **Mechanism Validation**: Are visual causal steps logically connected and complete?

## Evaluation Scale (1 to 5):
- **5 Perfect Target Match**: Generated image perfectly matches target description (and target image if available) with correct causal logic
- **4 High Quality Match**: Generated image achieves 80-90%+ of target requirements with only minor details missing or slightly incorrect; core causal changes are correct
- **3 Adequate Match**: Generated image meets basic requirements (60-70%) but has notable gaps, wrong aspects, or incomplete causal changes
- **2 Poor Match**: Generated image fails most target requirements (30-50% achievement) with major gaps or incorrect causal reasoning
- **1 Failed Match**: Generated image completely fails to match target or shows fundamental causal logic errors (<30% achievement)

### Example 1 (Score: 5): Perfect Potato Prevention
**Task**: "Apply lemon juice to prevent these cut potatoes from browning"
**Dimension**: "science"
**Keywords**: "citric acid, enzymatic browning, oxidation prevention"
**Target Description**: "cut potatoes remain white/pale after lemon juice application"

Figure 16: Prompt template for evaluating visual causal reasoning capabilities (RV). (Continued in Figure 17)

## Prompt for Reasoning Visual of Causal

**Evaluation**:
1. **Target Match**: ✓ All potatoes remain white/pale, ✓ Lemon juice clearly applied
2. **Visual Changes**: ✓ Lemon juice visible on potato surfaces, ✓ Potatoes maintain original color
3. **Domain Knowledge**: ✓ Citric acid prevention shown correctly, ✓ Application method appropriate
4. **Mechanism Validation**: ✓ Chemical prevention process visible, ✓ Complete coverage achieved

→ **reasoning_visual_score**: 5 (Perfect causal reasoning with complete prevention)

### Example 2 (Score: 3): Adequate Potato Prevention
**Task**: "Apply lemon juice to prevent these cut potatoes from browning"
**Dimension**: "science"
**Keywords**: "citric acid, enzymatic browning, oxidation prevention"
**Target Description**: "cut potatoes remain white/pale after lemon juice application"

**Evaluation**:
1. **Target Match**: ✗ Some potatoes show browning, ✗ Incomplete prevention
2. **Visual Changes**: ✓ Lemon juice partially applied, ✗ Some areas missed
3. **Domain Knowledge**: ✓ Basic citric acid concept shown, ✗ Application incomplete
4. **Mechanism Validation**: ✗ Chemical prevention partially failed, ✗ Coverage gaps

→ **reasoning_visual_score**: 3 (Adequate causal reasoning with partial prevention)

### Example 3 (Score: 1): Failed Potato Prevention
**Task**: "Apply lemon juice to prevent these cut potatoes from browning"
**Dimension**: "science"
**Keywords**: "citric acid, enzymatic browning, oxidation prevention"
**Target Description**: "cut potatoes remain white/pale after lemon juice application"

**Evaluation**:
1. **Target Match**: ✗ All potatoes heavily browned, ✗ No prevention visible
2. **Visual Changes**: ✗ No lemon juice visible, ✗ Potatoes completely oxidized
3. **Domain Knowledge**: ✗ No citric acid application shown, ✗ Wrong approach
4. **Mechanism Validation**: ✗ No chemical prevention, ✗ Complete failure

→ **reasoning_visual_score**: 1 (Complete failure of causal reasoning)

## Input
**Image 1: Original Image** (the starting point)
**Image 2: Generated Image** (the result after causal reasoning)
**Image 3: Target Image** (if available, the reference showing expected result)
**Task Instruction**: {prompt}
**Dimension**: {dimension}
**Keywords**: {keywords}
**Target Description**: {target_description}

## Output Format
{{
"reasoning_visual_score": X,
"reasoning": "1. Target Match 2. Visual Changes Analysis 3. Domain Knowledge Check 4. Mechanism Validation"
}}

Figure 17: Prompt template for evaluating visual causal reasoning capabilities (RV). (Continued from Figure 16)

**Prompt for Reasoning Alignment**

You are a professional AI evaluation specialist with expertise in causal reasoning assessment.

You will be given:

You are a professional AI evaluation specialist focusing on process-visual reasoning alignment assessment.

You will be given:
1. **Original Image**: the starting point
2. **Generated Image**: the reasoning result
3. **Task Instruction**: what reasoning should be performed
4. **Think Output**: the reasoning process text generated by the model

Your Objective:
Evaluate whether the **reasoning process text** and the **visual reasoning result** are aligned and consistent with each other.
Focus on whether what the model thought and what the model visually produced match.

## Alignment Evaluation Criteria:
- **Process-Visual Consistency**: Do the written reasoning steps match the visual changes?
- **Conclusion Coherence**: Do text conclusions align with visual outcomes?
- **Step-by-Step Alignment**: Does each reasoning step in text correspond to visual evidence?
- **Logical Consistency**: Are there contradictions between thought process and visual result?

## Domain-Specific Considerations:
- **Science**: Apply scientific principles and natural laws; verify that reasoning follows established scientific facts and theories; reject unscientific claims or impossible phenomena
- **Humanity**: Consider cultural, historical, and social contexts; ensure reasoning respects cultural norms and historical accuracy; avoid cultural insensitivity or anachronisms
- **Common Sense**: Use everyday knowledge and practical understanding; verify reasoning aligns with real-world experience and logical expectations; reject unrealistic or impractical scenarios
- **Logic**: Follow formal reasoning and mathematical principles; ensure logical consistency and mathematical accuracy; reject logical fallacies or mathematical errors

## Key Questions:
1. **Does the visual result reflect the written reasoning?** Are the visual changes consistent with what was described in the think output?
2. **Are the conclusions aligned?** Do both process and visual reasoning reach the same conclusions?
3. **Is the reasoning coherent?** Are there contradictions between what was thought and what was visually produced?
4. **Is the task prompt correctly understood?** Do both the process text and visual result demonstrate correct understanding of what the task is asking for?

## Evaluation Scale (1 to 5):
- **5 Perfect Alignment**: Process text and visual result are **completely consistent** and mutually supporting with **zero contradictions**; all process claims match visual evidence exactly; AND both correctly understand and implement the task prompt
- **4 High Quality Alignment**: Process and visual achieve 80-90%+ alignment with only minor inconsistencies that don't affect core reasoning; AND both generally follow the task prompt correctly
- **3 Adequate Alignment**: Some alignment present (60-70%) but clear discrepancies between process and visual reasoning; notable inconsistencies exist; OR good internal alignment but significant misunderstanding of task prompt
- **2 Poor Alignment**: Minimal alignment (30-50%) with major contradictions between written process and visual result; significant mismatches; OR both process and visual fundamentally misunderstand the prompt
- **1 No Alignment**: Process text and visual result are contradictory or completely unrelated (<30% alignment); OR complete failure to understand task prompt

Figure 18: Prompt template for evaluating reasoning alignment capabilities (Align.). (Continued in Figure 19)

> **Prompt for Reasoning Alignment**
>
> ## CRITICAL ALIGNMENT CONSTRAINT:
> **Alignment score cannot exceed visual reasoning score by more than 1 point.**
> - If visual reasoning = 1, alignment can be at most 2
> - If visual reasoning = 2, alignment can be at most 3
> - If visual reasoning = 3, alignment can be at most 4
> - If visual reasoning = 4-5, alignment can be 4-5
>
> This ensures logical consistency: you cannot have high alignment with poor visual reasoning.
>
> ## Reasoning Steps:
> 1. **Extract Process Claims**: What does the think output claim will happen or should be done?
> 2. **Identify Visual Evidence**: What changes are actually visible in the generated image?
> 3. **Compare Alignment**: Do the process claims match the visual evidence?
> 4. **Assess Consistency**: Are there any contradictions between thought and visual result?
> 5. **Evaluate Prompt Understanding**: Do both the process text and visual result correctly understand and implement the task prompt requirements?
> 6. **Domain Knowledge Check**: Do both process and visual reasoning follow domain-specific scientific/cultural/commonsense/logical principles? Ensure alignment respects established domain knowledge and reject violations of domain principles
> 7. **Apply Alignment Constraint**: Ensure alignment score does not exceed visual reasoning quality by more than 1 point
>
> ## Input
> **Image 1: Original Image** (the starting point)
> **Image 2: Generated Image** (the reasoning result)
> **Task Instruction**: {prompt}
> **Think Output**: {think_output}
>
> ## Output Format
> {{
> "reasoning_alignment_score": X,
> "reasoning": "1. Process Claims 2. Visual Evidence 3. Alignment Comparison 4. Consistency Assessment 5. Prompt Understanding 6. Domain Knowledge Check 7. Alignment Constraint"
> }}

Figure 19: Prompt template for evaluating reasoning alignment capabilities (Align.). (Continued from Figure 18)

**Prompt for Reasoning Visual Consistency**

You are a professional visual evaluation specialist focusing on image consistency assessment.

You will be given:
1. **Original Image**: the starting point
2. **Generated Image**: the result after reasoning/editing
3. **Task Instruction**: the reasoning or editing task performed

Your Objective:
Evaluate whether **non-target elements** in the generated image remain **visually consistent** with the original image. Focus exclusively on elements that should NOT have changed according to the task instruction.

## Consistency Evaluation Guidelines:

### Elements to Preserve:
- **Background Elements**: Scenery, environment, setting details not mentioned in task
- **Unrelated Objects**: Items not involved in the reasoning/editing process
- **Structural Elements**: Basic composition, layout, perspective (unless task requires change)
- **Identity Preservation**: People, animals, or objects should maintain their core identity
- **Style Consistency**: Overall visual style, lighting conditions, color palette

### Elements That May Change (Task-Dependent):
- **Target Objects**: Items explicitly mentioned in the task instruction
- **Direct Consequences**: Changes that logically result from the intended transformation
- **Process Effects**: Visual effects directly caused by the reasoning process

## Evaluation Scale (1 to 5):
- **5 Perfect Consistency**: All non-target elements remain **visually identical** to original with **zero unintended changes**; perfect preservation of all non-instructed elements
- **4 Minor Inconsistency**: **Minimal unintended changes** that are barely noticeable and don't affect coherence; only very small discrepancies
- **3 Noticeable Inconsistency**: **Clear unintended changes** in background or unrelated elements that affect coherence; notable inconsistencies exist
- **2 Significant Inconsistency**: **Multiple unintended changes** that significantly compromise visual coherence; major inconsistencies
- **1 Severe Inconsistency**: **Major unintended alterations** that make image appear largely different; fundamental consistency breakdown

## Reasoning Steps:
1. **Identify Target Elements**: What elements should change according to the task?
2. **Isolate Preserve Elements**: What elements should remain unchanged?
3. **Compare Preservation**: Are the preserve elements visually consistent with original?
4. **Assess Impact**: How do any inconsistencies affect overall visual coherence?

## Input
**Image 1: Original Image** (the starting point)
**Image 2: Generated Image** (the result after reasoning/editing)
**Task Instruction**: {prompt}

## Output Format
{{
"visual_consistency_score": X,
"reasoning": "1. Target Elements 2. Preserve Elements 3. Preservation Comparison 4. Impact Assessment"
}}

Figure 20: Prompt template for evaluating visual consistency (VC.). (Continued from Figure 20)

Prompt for Image Quality

You are a professional image quality assessor specializing in AI-generated content evaluation.

You will be given:
1. **Generated Image**: an AI-generated image to evaluate

Your Objective:
Evaluate the **perceptual quality** of the AI-generated image, focusing on technical excellence, visual coherence, and absence of generation artifacts.

## Quality Assessment Dimensions:

### Structural Coherence
- **Anatomy/Geometry**: Correct proportions, realistic structures, proper object shapes
- **Spatial Relationships**: Logical positioning, appropriate scale relationships
- **Compositional Logic**: Coherent scene layout, proper perspective

### Visual Fidelity
- **Texture Quality**: Realistic surface textures, appropriate material appearance
- **Detail Clarity**: Sharp important details, appropriate level of detail throughout
- **Color Accuracy**: Natural color distribution, proper lighting/shadow

### Generation Artifacts
- **Duplication Issues**: Repeated elements, phantom objects, merged features
- **Blending Problems**: Unnatural transitions, ghosting effects, edge artifacts
- **Distortion Errors**: Warped features, impossible geometries, scale inconsistencies

### Overall Naturalness
- **Photorealism**: Does the image look natural and believable?
- **Coherent Style**: Consistent visual style throughout the image
- **Professional Quality**: Would this pass as high-quality content?

## Evaluation Scale (1 to 5):
- **5 Excellent Quality**: **Professional-grade image** with **no noticeable artifacts or flaws**; perfect technical excellence and photorealistic quality
- **4 Good Quality**: **High-quality image** with **one minor flaw** that doesn't affect overall impression; minimal quality issues
- **3 Acceptable Quality**: **Decent image** with **some noticeable flaws** but overall usable; clear quality problems exist
- **2 Poor Quality**: **Multiple significant flaws** that detract from image usability; major quality problems
- **1 Very Poor Quality**: **Major structural problems**, severe artifacts, unusable quality; fundamental quality breakdown

## Quality Checklist:
For each dimension, mark ✓ (satisfactory) or ✗ (problematic):
- Structural coherence: ✓/✗
- Visual fidelity: ✓/✗
- Artifact-free: ✓/✗
- Overall naturalness: ✓/✗

## Reasoning Steps:
1. **Structural Analysis**: Assess geometric and anatomical correctness
2. **Fidelity Evaluation**: Check texture, detail, and color quality
3. **Artifact Detection**: Identify any generation artifacts or distortions
4. **Naturalness Assessment**: Evaluate overall believability and professional quality

## Input
**Generated Image**

## Output Format
{{
"image_quality_score": X,
"reasoning": "1. Structural Analysis 2. Fidelity Evaluation 3. Artifact Detection 4. Naturalness Assessment"
}}

Figure 21: Prompt template for evaluating image quality (IQ.). (Continued from Figure 21)

**World Model for Embodied Task**

```
SYSTEM_PROMPT = '''You are a robotics trajectory planner with
    visualization capabilities. When given a robotics scene, you
    must:

1. Generate a trajectory visualization image:
    - Overlay 10 waypoint markers on the input scene
    - Style: Blue circles with white outlines + connecting
        trajectory lines
    - Labels: 'traj1', 'traj2', ..., 'traj10'
    - Reference the example image (Image 2) for exact visualization
        style

2. Output pixel coordinates based on your visualization:
    - Format: [[x1, y1], [x2, y2], ..., [x10, y10]]
    - Coordinate system: (0, 0) at top-left corner

Constraints:
- Start from current end-effector position
- End at task completion position
- Generate smooth, collision-free waypoints
- All coordinates must be within image boundaries

Format your output as:
<think>
[Your analysis and planning process]
</think>

Final Answer: [Trajectory coordinates: [[x1, y1], [x2, y2], ..., [
    x10, y10]]]'''
```

**World Model for Physical Task**

```
SYSTEM_PROMPT = '''You are a physics simulation AI with image
    generation capabilities. Given a physical scenario, you must:

1. Analyze the initial scene (Image 1):
- What objects are present? (cars, balls, liquids, blocks, pulleys,
    etc.)
- What are their initial positions and states?
- What forces or motions will be applied?

2. Use Image 2 (if provided):
- It may show additional context of the scene

3. Generate a physics simulation result image:
Simulate what happens when physics is applied and generate the
    resulting scene:
- Apply gravity, momentum, friction, collision dynamics
- Simulate the complete physical process (cars moving and colliding
    , balls falling, liquids flowing, pulleys rotating)
- Generate an image showing the final state or outcome after
    physics simulation
- The simulated image should naturally show where objects end up,
    what gets affected, and what the result is

Format your output as:
<think>
1. Initial state: [describe the setup]
2. Generate the physics simulation image
3. Analyze the generated image: [what does the simulation show?]
4. Determine answer: [based on the generated image, what is the
    result?]
</think>

Final Answer: [exact answer format requested in question]'''
```

**Logic & Math**

```
SYSTEM_PROMPT = '''You are a helpful AI assistant. You need to
    think about the given prompt/question and any hints provided,
    then generate USEFUL VISUAL AIDS based on the hints during your
    thinking process, and finally answer the question based on your
    analysis and the generated images.

IMPORTANT REQUIREMENTS:
1. You MUST generate images that are USEFUL VISUAL AIDS for solving
    the problem (e.g., with auxiliary lines, labels, annotations,
    constructions that help solve the problem)
2. Do NOT generate images that merely replicate the given figure
    without adding helpful information
3. You MUST provide a final answer after your thinking process

Enclose your thinking process within <think> </think> tags,
    generate relevant images during thinking, then provide your
    final answer.

Format your output as:
<think>
Step 1: Analyze what auxiliary constructions would help solve this
    problem.
Step 2: Generate the visual aid with those constructions.
[generate USEFUL images with helpful additions like auxiliary lines
    , labels, or constructions]
Step 3: OBSERVE the generated image carefully and use the visual
    information to perform your reasoning.
Step 4: Based on what you see in the generated image, work through
    the solution.
</think>

Final Answer: [your answer based on the generated images and
    analysis]

REMEMBER: The generated images must be USEFUL VISUAL AIDS that add
    value beyond the original figure.'''
```

**Visual Perception for Jigsaw**

```
SYSTEM_PROMPT = '''You are a helpful AI assistant solving visual
    jigsaw puzzles. You need to analyze the puzzle image with a gray
     box covering part of it, then generate a completed image by
    filling in the missing area, and finally select the correct
    option.

IMPORTANT REQUIREMENTS:
1. You MUST first generate a completed image by filling in the
    missing area covered by the gray box
2. Compare your generated full image with the original puzzle to
    validate consistency
3. Use the generated complete image to determine which option (A, B
    , C, or D) correctly fills the missing area
4. Do NOT answer before generating the completed image

Enclose your thinking process within <think> </think> tags,
    generate the completed image during thinking, then provide your
    final answer.

Format your output as:
<think>
Step 1: Analyze what is visible in the puzzle and what patterns/
    objects might be in the missing area.
Step 2: Generate a completed image by filling in the missing top
    part of the puzzle.
[Generate the full completed image]
Step 3: Carefully observe your generated complete image and compare
     it with the original puzzle to ensure consistency.
Step 4: Compare your generated complete image with each option (A,
    B, C, D) to find which one matches the missing area in your
    generated image.
</think>

Final Answer: [A/B/C/D]'''
```

**Visual Perception for Multi-view Reasoning**

```
SYSTEM_PROMPT = '''You are a helpful AI assistant analyzing multi-
    view images to determine camera movement direction. Given two
    images taken from different camera positions around the same
    scene, you need to reason about the spatial relationships and
    determine the camera rotation direction.

IMPORTANT REQUIREMENTS:
1. You MUST first generate a wider-angle image taken from farther
    away that includes ALL objects visible in both Image 1 and Image
     2
2. This generated image should be like stepping back and using a
    wider lens - showing more of the scene in one frame
3. Use this wider-angle view to understand the spatial relationship
     between the two camera positions
4. Determine if the camera rotated counterclockwise (left) or
    clockwise (right) from Image 1 to Image 2
5. Do NOT answer before generating the wider-angle image

Think of it like this: If you step back from the scene and take a
    photo with a wider-angle lens, you can see all the objects from
    both viewpoints in one image.

Enclose your thinking process within <think> </think> tags,
    generate the wider-angle image during thinking, then provide
    your final answer.

Format your output as:
<think>
Step 1: Identify all objects visible in Image 1 and Image 2.
Step 2: Generate a wider-angle image from farther away that
    includes all these objects.
[Generate the wider-angle image showing the complete scene]
Step 3: Use this wider view to understand where the two cameras
    were positioned.
Step 4: Determine the rotation direction: counterclockwise (left)
    or clockwise (right)?
</think>

Final Answer: [left/right]'''
```

