# OpenReview forum: "ROVER: Benchmarking Reciprocal Cross-Modal Reasoning for Omnimodal Generation"
_ICLR.cc/2026/Conference — ICLR 2026 Poster_

### Official Review · Reviewer_Bjyh · 2025-10-14

**Soundness:** 2
**Presentation:** 4
**Contribution:** 3
**Rating:** 4
**Confidence:** 4

**Summary:**

This paper introduces ROVER, a benchmark designed to evaluate the reciprocal cross-modal reasoning capabilities of Unified Multimodal Models (UMMs). The authors argue that existing benchmarks fail to assess how models use one modality to guide or verify outputs in another, instead testing text and image abilities in isolation. ROVER addresses this gap with over 1,200 tasks that require integrated reasoning across modalities, focusing on verbally-augmented reasoning for visual generation and visually-augmented reasoning for verbal generation. By testing 17 state-of-the-art UMMs, the study finds that cross-modal reasoning skills strongly correlate with visual generation performance. However, it also reveals that current models are severely limited in visually-augmented reasoning, showing particular weakness in logical tasks compared to perception and physical modeling.

**Strengths:**

The paper introduces the first benchmark that requires generating both visual and textual content for joint visual and textual reasoning, effectively unifying the two modalities. The authors conduct extensive experiments demonstrating that incorporating multimodal generation improves performance compared to text-only generation during evaluation. The finding that models with stronger image–text interleaving capabilities outperform image-editing models is also noteworthy. Overall, the paper provides clear evidence that text generation supports image generation, and image generation, in turn, enhances textual reasoning.

**Weaknesses:**

1. The paper's evaluation methodology relies heavily on the quality of generated images, particularly for Reasoning Visual (RV), which requires generating coherent images to facilitate correct reasoning. However, the use of a VLM as a judge is questionable. Figure 8 reveals a significantly low correlation (0.63) and a high MAE of nearly 1.0 between GPT's evaluations and human judgments. Assuming the four human evaluators provide a more reliable gold standard, this discrepancy undermines the validity of using VLMs to assess RV quality. This concern is amplified by prior work demonstrating that even state-of-the-art VLMs struggle with fundamental spatial and temporal reasoning.
2. In addition to the questionable reliability of the VLM-as-a-judge paradigm, the paper fails to address the financial costs associated with using the GPT for evaluation, a notable omission given the large volume of generated images involved.
3. While the authors present Figure 7b to show the correlation between different task types, this finding is largely unsurprising due to the semantic definitions of the tasks. Crucially, it remains unclear how this analysis provides actionable insights or how it might guide the development of future models.

**Questions:**

See weaknesses.

---

> ### Author Response · Authors · 2025-11-17
> **Rebuttal by Authors (1/2)**
>
> > **Q1**: *The paper's evaluation methodology relies heavily on the quality of generated images, particularly for Reasoning Visual (RV), which requires generating coherent images to facilitate correct reasoning. However, the use of a VLM as a judge is questionable.*
>
> **A1**: Thank you for raising this important problem!
> - **(1.1)** We would like to clarify this potential confusion: as stated in `L214`, the Reasoning Visual (RV) metric evaluates **how well the final generated image reflects the target description**, rather than assessing the model's internal reasoning quality.
> - **(1.2)** If the concern instead refers to the Interleaved Reasoning (IR) metric, the evaluation details, using our ground-truth visual CoT images and the human validation analysis, are provided in `Appendix B` and `C`.
> - **(1.3)** GPT-based evaluation has become the standard and well-validated protocol in recent visual generation and editing benchmarks (e.g., KRIS-Bench [1], RISEBench [2], WISE [3], IntelligentBench [4], GEditBench [5]), and we adopt the same protocol for comparability and consistency, as it remains **the most reliable assessment for complex multimodal generation**.
>
> ---
>
> > **Q2**: *Figure 8 reveals a significantly low correlation (0.63) and a high MAE of nearly 1.0 between GPT's evaluations and human judgments. Assuming the four human evaluators provide a more reliable gold standard, this discrepancy undermines the validity of using VLMs to assess RV quality.*
>
> **A2**: Thank you for the insightful question!
> - **(2.1)** The lower correlations in our initial RV reliability study were mainly due to limited score variance. Among 200 samples from three models (Nano-Banana, GPT-5, BAGEL-Think), more than 30% outputs fell in the mid-range (3–4). Human experts tended to choose 3, while GPT-4.1 was slightly more lenient and chose 4, which reduces the linear correlation, leading to a higher MAE, even though **GPT-4.1 still reliably distinguishes high-quality samples and low-quality ones**.
> - **(2.2)** To further strengthen the reliability analysis, we expanded our study to 8 human judges for 10 models (1000 instances) with larger score variance and diverse samples, and updated the results in `Figure 8`. RV now achieves **r = 0.75 and MAE = 0.69**, which are comparable to or better than prior benchmarks [1,2]. Importantly, **GPT-based RV scores yield the same model ranking as human scores**, supporting the reliability of our evaluation protocol.
>
> ---
>
> > **Q3**: *In addition to the questionable reliability of the VLM-as-a-judge paradigm, the paper fails to address the financial costs associated with using the GPT for evaluation, a notable omission given the large volume of generated images involved.*
>
> **A3**: Thank you for the question!
> - **(3.1)** As discussed earlier, our evaluation protocol follows the standard practice established in recent visual-generation and editing benchmarks, which also rely on GPT-based judges.
> - **(3.2)** To examine whether our results remain stable under a more cost-efficient and fully open setup, we further assessed an open-source evaluator. We used the newly released Qwen3-VL-32B-Thinking to evaluate generated images from 10 models, 1000 instances and compared its agreement with human evaluations against GPT-4.1. The results (Pearson r and MAE) are summarized below:
>
> | Judge (Average on ROVER-IG)   | Correlation (r) ↑ | MAE ↓ |
> |---------------------------|-------------|--------|
> | **GPT-4.1 (Ours)**               | 0.81        | 0.58   |
> | **Qwen3-VL-32B-Thinking** | 0.74        | 0.68   |
>
> - **(3.3)** Although the open-source judge lags slightly behind GPT-4.1, **it still aligns well with human ratings**, showing that **our benchmark remains reliable under both proprietary and open-source evaluators**. We also agree that exploring more cost-efficient and open evaluation pipelines is a valuable future direction.

---

> ### Author Response · Authors · 2025-11-17
> **Rebuttal by Authors (2/2)**
>
> > **Q4**: *While the authors present Figure 7b to show the correlation between different task types, this finding is largely unsurprising due to the semantic definitions of the tasks. Crucially, it remains unclear how this analysis provides actionable insights or how it might guide the development of future models.*
>
> **A4**: Thank you for this insightful question!
> - **(4.1)** The purpose of `Figure 7b` is to examine how current UMMs internally structure different forms of reasoning. **The observed correlations are not predetermined by task semantics**: even semantically distinct tasks such as quantitative and causal/spatial reasoning exhibit high coherence. This indicates **an emergent coupling among perceptual and physically grounded reasoning types that is not obvious from task definitions alone**.
> - **(4.2)** In contrast, symbolic reasoning show consistently weak correlation with these perceptual categories, revealing **a systematic gap in how UMMs connect symbolic abstraction with visually grounded reasoning**. This pattern is also consistent with our ROVER-TG findings and points to a concrete direction for future model development. **Improving the integration between symbolic reasoning and perceptual reasoning remains an open challenge**.
>
> ---
>
> **References:**
>
> *[1] KRIS-Bench: Benchmarking Next-Level Intelligent Image Editing Models. Wu et al. NeurIPS 2025.*
>
> *[2] Envisioning Beyond the Pixels: Benchmarking Reasoning-Informed Visual Editing. Zhao et al. NeurIPS 2025.*
>
> *[3] WISE: A World Knowledge-Informed Semantic Evaluation for Text-to-Image Generation. Niu et al. arxiv 2025.*
>
> *[4] Emerging Properties in Unified Multimodal Pretraining. Deng et al. arxiv 2025.*
>
> *[5] Step1x-edit: A Practical Framework for General Image Editing. Liu et al. arXiv 2025*
>
> ---
>
> If you have any additional concerns, please do not hesitate to let us know. We are more than willing to address them and sincerely appreciate your valuable feedback and support.

---

> > ### Comment · Reviewer_Bjyh · 2025-11-17
> >
> > I appreciate the authors' quick rebuttal and it has resolved most of my concerns. I have one final question: what do the authors vision a benchmark that integrates symbolic and perceptual reasoning together as? For instance, what would an example that is "spatial" and "abstract" or "temporal" and "mathematical" be, if I understood the meaning of correlation correctly?

---

> > > ### Author Response · Authors · 2025-11-18
> > > **Official Comment by Authors**
> > >
> > > Thank you for your timely response and for raising this insightful follow-up question.
> > > To illustrate what integrated symbolic–perceptual tasks may look like, we offer two simple but concrete examples:
> > >
> > > * **(1) Spatial + Abstract** example:
> > >
> > > Given an image containing multiple blocks and other objects in a scene, identify which objects violate the rule: "Red blocks must only be placed on blue blocks, and blue blocks must only be placed on green blocks."
> > >
> > > This requires the model to jointly ground **visual perception** (object identity, color classification, and block-on-block spatial relations) with an **abstract symbolic constraint**.
> > >
> > > * **(2) Temporal + Mathematical** example:
> > >
> > > Given a video shows a ball bouncing with decreasing height, predict the height of the 6th bounce assuming the decay pattern continues.
> > >
> > > Here, the model must extract **temporal events** from raw video **(perceptual reasoning)**, **infer the underlying decay pattern, and extrapolate the next value (mathematical)**.
> > >
> > > These examples illustrate scenarios in which symbolic and perceptual reasoning must work together, clarifying the type of integrated reasoning that remains challenging for current models. We hope this helps convey the insight for future multimodal reasoning benchmarks.
> > >
> > > We sincerely appreciate your valuable feedback, which has significantly contributed to the improvement of our work!

---

> > > > ### Comment · Reviewer_Bjyh · 2025-11-18
> > > >
> > > > I again appreciate the timely feedback, and I have raised my score to 6.

---

> > > > > ### Author Response · Authors · 2025-11-18
> > > > > **Thank you for your acknowledgment!**
> > > > >
> > > > > Dear Reviewer `Bjyh`,
> > > > >
> > > > > We sincerely appreciate your acknowledgment of our rebuttal. We are glad to hear that your concerns have been addressed and are voting toward acceptance.
> > > > >
> > > > > Your support and encouragement mean a lot to us!
> > > > >
> > > > > Best regards,
> > > > >
> > > > > The Authors of Submission 1795

---

### Official Review · Reviewer_icDF · 2025-10-19

**Soundness:** 3
**Presentation:** 3
**Contribution:** 3
**Rating:** 6
**Confidence:** 5

**Summary:**

This paper proposes ROVER, a benchmark with over 1,200 tasks and 2,048 images for reciprocal cross-modal reasoning. ROVER has two parts: ROVER-IG (language guiding image generation) and ROVER-TG (vision aiding text generation). They tested 17 UMMs with a VLM judge plus expert checks, finding cross-modal reasoning ties to visual generation, but models struggle with vision-aided logic tasks.

**Strengths:**

1. This paper is generally well-written and easy to follow, with clearly illustrated figures.
2. ROVER covers a wide range of both language-reasoning tasks and visual-reasoning tasks, and uses a comprehensive evaluation method (VLM + expert validation) to ensure reliability.
3. The authors evaluate 17 unified multimodal models and provide insightful findings.

**Weaknesses:**

1. The benchmark heavily depends on a "VLM-as-a-judge" for scoring complex reasoning qualities. The paper's own user study (Figure 8) shows that while correlation is good, there are noticeable discrepancies, especially for reasoning-related metrics. This introduces a potential bias, where the benchmark might favor models whose outputs align with the judging VLM's own reasoning patterns.
2. As listed in Table 3, language-only models often match or exceed the performance of unified models on reasoning tasks, questioning whether the current task design truly requires cross-modal reasoning for optimal results ("thinking with images").​

**Questions:**

1. How does the up-to-date Gemini-2.5-pro perform on this benchmark?

---

> ### Author Response · Authors · 2025-11-17
> **Rebuttal by Authors**
>
> > **Q1**: *The benchmark heavily depends on a "VLM-as-a-judge" for scoring complex reasoning qualities. The paper's own user study (Figure 8) shows that while correlation is good, there are noticeable discrepancies, especially for reasoning-related metrics. This introduces a potential bias, where the benchmark might favor models whose outputs align with the judging VLM's own reasoning patterns.*
>
> **A1**: Thank you for this valuable question!
> - **(1.1)** GPT-based evaluation has become the standard and well-validated protocol in recent visual generation and editing benchmarks (e.g., KRIS-Bench [1], RISEBench [2], WISE [3], IntelligentBench [4], GEditBench [5]), and we adopt the same protocol for comparability and consistency, as it remains **the most reliable assessment for complex multimodal generation**.
> - **(1.2)** Evaluating the quality of chain-of-thought (CoT) reasoning remains challenging even for the latest LLM/VLM judges, and our multi-dimensional metrics are designed to **emphasize whether the reasoning process contributes to the final outcome** rather than to assess CoT quality in isolation, as detailed in our evaluation prompts (`Appendix E.1`). The reasoning-process scores serve as supplementary metrics for analysis, and we also agree that **developing more robust cross-modal reasoning evaluation methods is a promising direction for future work**.
>
> ---
>
> > **Q2**: *As listed in Table 3, language-only models often match or exceed the performance of unified models on reasoning tasks, questioning whether the current task design truly requires cross-modal reasoning for optimal results ("thinking with images").*
>
> **A2**: Thank you for this constructive feedback!
> - **(2.1)** We have added a more detailed description of the data curation process for TG about visual reasoning steps in `Appendix B` for further clarification. Specifically:
>    - For logical problems, we curated over 1,000 instances with ground-truth visual annotations. We performed an automatic sanity check using GPT-5 and selected 150 cases where **the presence of the ground-truth visual CoT changes the model's answer**, indicating that **visual reasoning steps have a concrete impact on model behavior**.
>    - For visual and physical reasoning tasks, our task definitions inherently require intermediate visual cues. Examples include robot arm trajectories in manipulation tasks, complete reference images in jigsaw tasks, and wide-view images that support spatial understanding in multi-view spatial reasoning. These intermediate visual cues encode **fine-grained geometric and perceptual structure that is hard to capture through text-only reasoning**.
> - **(2.2)** The relatively small gains of current unified models reflect **their limited ability to generate accurate visual reasoning steps**, as illustrated in `Figure 5` and the examples provided on our project page, **rather than from the tasks not requiring cross-modal reasoning**.
>
> ---
>
> > **Q3**: *How does the up-to-date Gemini-2.5-pro perform on this benchmark?*
>
> **A3**: Thank you for this question!
> - **(3.1)** The "Nano-Banana" model in our benchmark corresponds to **Gemini-2.5-Flash-Image** [6], which is the up-to-date image generation model from Google. We have extensively evaluated this model throughout the paper and have added its experiment setup in `Section 4.1` for clarity.
> - **(3.2)** We note that **Gemini-2.5-Pro does not provide an image-generation interface**, so it cannot be directly evaluated on our benchmark.
>
> ---
>
> **Reference:**
>
> *[1] Wu et al. KRIS-Bench: Benchmarking Next-Level Intelligent Image Editing Models. NeurIPS 2025.*
>
> *[2] Zhao et al. Envisioning Beyond the Pixels: Benchmarking Reasoning-Informed Visual Editing. NeurIPS 2025.*
>
> *[3] Niu et al. WISE: A World Knowledge-Informed Semantic Evaluation for Text-to-Image Generation. arxiv 2025.*
>
> *[4] Deng et al. Emerging Properties in Unified Multimodal Pretraining. arxiv 2025.*
>
> *[5] Liu et al. Step1x-edit: A Practical Framework for General Image Editing. arXiv 2025.*
>
> *[6] https://aistudio.google.com/models/gemini-2-5-flash-image*
>
>
> ---
>
> If there are any remaining concerns, we would be happy to address them. Thank you again for your thoughtful and constructive feedback!

---

> ### Author Response · Authors · 2025-11-26
> **Gentle Reminder: Discussion Phase Closing Soon**
>
> Dear Reviewer `icDF`,
>
> Thank you very much for your thoughtful and constructive review of our submission.
>
> Following your suggestions, we validated the VLM-based evaluation as a standard protocol and clarified its focus on reasoning contribution. We demonstrated the necessity of visual reasoning by detailing the data curation in `Appendix B` and showing how visual steps actively impact model outputs. We also clarified the model selection in `Section 4.1`.
>
> As the rebuttal period draws to a close, we kindly ask if our responses have fully addressed your questions. If our responses have satisfactorily resolved your concerns, we would be deeply grateful for a reconsideration of your rating. Should any issues remain, we are more than willing to provide further clarifications to address them comprehensively.
>
> We truly appreciate your time and effort in reviewing our work and for your suggestions that have helped strengthen our manuscript.
>
> Warm regards,
>
> The Authors of Submission 1795

---

> > ### Comment · Reviewer_icDF · 2025-11-28
> >
> > Thanks for the response. I believe my original rating is reasonable.

---

### Official Review · Reviewer_FBtf · 2025-10-30

**Soundness:** 1
**Presentation:** 2
**Contribution:** 4
**Rating:** 4
**Confidence:** 4

**Summary:**

This paper introduces ROVER, a new benchmark designed to evaluate reciprocal cross-modal reasoning in unified multimodal models (UMMs), i.e., the ability to use one modality (text or image) to guide reasoning and generation in the other. Existing evaluations tend to isolate modalities, emphasizing either textual or visual reasoning in isolation, which fails to capture the intended integration of modern UMMs.

ROVER fills this gap through over 1,200 human-annotated tasks grounded in 2,048 images, spanning two complementary settings: (1) verbally-augmented reasoning for visual generation, where structured verbal reasoning guides faithful image synthesis, and (2) visually-augmented reasoning for verbal generation, where models generate intermediate visualizations to support their reasoning.

The authors evaluate 17 state-of-the-art UMMs and find that cross-modal reasoning ability correlates with visual generation performance, especially for interleaved text–image tasks. However, most models remain weak in visually-augmented reasoning, particularly in logical reasoning scenarios.

Overall, the work is a good first step towards analyzing cross model reasoning abilities of current UMMs and avenues of improvement, but still requires more work to solidify its usability and interpretability.

**Strengths:**

1. Importance of cross model reasoning in UMM and problem formulation in two complementary settings of verbally-augmented reasoning for visual generation (ROVER-IG) and visually-augmented reasoning for verbal generation (ROVER-TG) is interesting, useful and novel.
2. Careful dataset design into top level domains and subtasks for both ROVER-IG and ROVER-TG
3. Detailed metrics that aim to provide a holisitic understanding of the model performance in either settings.
4. Interesting analysis like coherence between reasoning substasks.

**Weaknesses:**

The paper gives a good shot to cover a novel perspective but falls short in these following areas:

1. Stretch / Over claims:
a) "Pg 5 section 4.1 (last para) the authors claim that gaps in reasoning process and alignment is the fundamental driver of diminished visual generation performance" but as seen for table 2, if you look at natural science or logic for instance for both closed and open source model, similar RP and align scores show great variability in RV scores.
b) "Pg 7 section 4.2 Models demonstrate superior interleaved reasoning performance on physical world and visual perception tasks compared to logical reasoning challenges" is not supported in table 3, model perform similarly for the best for visual perception only and they have similar low performance in logic and physical world domains.

2. Clarifications
a). It is difficult to infer anything from the % reported in the paper, none of them mention if its absolute, or relative and relative with respect to what ?
b) visual generation performance on pg 5 last paragraph is vague. from reading context, i can map it to RV but would urge the authors to make explicit connections between numbers and metrics, especially when they define them
c) Section 4.3 Cross-modal Reasoning matters for UMMs: Could not follow through this analysis, CLIP-1 and edit world are introduced out of the blue without prior context. Fig on pg 9 top right has the corresponding details but is not reference in text and the figure itself is unclear, with some bars having +ve/-ve value and being of different lengths + no caption. Could not make sense of this at all

3. Judge reliability evaluation
a) Human correlation of RV one of the important metrics for IG is low
b) Only IG metrics undergo reliability evaluation what about TG metrics which are also llm judges ?
c) The models used for judge calibration are either closed source models or the strongest open-source model. This could potentially add bias in score calibration, having a weaker open source model being part of calibration can ensure that the entire spectrum of scores is callibrated.

4. Missing details
a) Fig 4, it would be nice to see the reasoning generated by the models in addition to input text and generated image, to better analyze reasoning alignment.
b) Table 2 does not clearly indicate which models are interleaved vs single turn, image editing only vs UMM but uses these terminologies in the analysis section when table 2 is referenced. Appendix does provide some insight to them but still models like Show-o2, Blip3o-8b, Janus-Pro-7b, etc have not been classified, making it difficult to relate to the claims in the paper

5. Overall the paper writing needs to improve, better references to figures and tables, improved captions

**Questions:**

I have added my questions as part of weakness itself, would urge the authors to respond to them. In its current state the work does not merit publication, however if the authors make the necessary clarifications and substantiate their claims, the benchmark would be more useful and i can consider bumping my score.

---

> ### Author Response · Authors · 2025-11-17
> **Rebuttal by Authors (1/3)**
>
> Thank you for your thoughtful and positive feedback regarding our problem formulation, dataset design, and analysis. We address your concerns point by point.
>
> ---
>
> > **Q1(a)**: *"Pg 5 section 4.1 (last para) the authors claim that gaps in reasoning process and alignment is the fundamental driver of diminished visual generation performance" but as seen for table 2, if you look at natural science or logic for instance for both closed and open source model, similar RP and align scores show great variability in RV scores.*
>
> **A1(a)**: Thank you for your constructive feedback!
> - **(1a.1)** As shown in the Overall column of `Table 2`, open-source models have noticeably lower RP and Alignment than closed-source models and accordingly obtain much lower RV. This overall pattern supports that these factors are strongly related to final visual generation (RV) performance.
> - **(1a.2)** We also note that RP, Alignment, and RV are evaluated with different prompts and protocols, so their absolute values are not directly comparable; only their relative ordering across models is meaningful.
> - **(1a.3)** Following your suggestion, we have refined the wording in `Section 4.2`, `L395-400` to provide a more precise description of this relationship, using "strong contributors" rather than "fundamental driver." This revision offers a clearer and more accurate interpretation of the results in `Table 2`.
>
> ---
>
> > **Q1(b)**: *Pg 7 section 4.2 Models demonstrate superior interleaved reasoning performance on physical world and visual perception tasks compared to logical reasoning challenges" is not supported in table 3, model perform similarly for the best for visual perception only and they have similar low performance in logic and physical world domains.*
>
> **A1(b)**: Thank you for the helpful question!
> - **(1b.1)** We would like to clarify that the raw IR scores in `Table 3` should not be directly compared across the three domains. Each domain employs different IR and Alignment prompts that focus on different parts of the reasoning process, as explained in `Appendix E.1`.
> - **(1b.2)** Our original statement referred to the observation that models, especially closed-source models, show small but consistent accuracy improvements over text-only reasoning on physical-world tasks, whereas logical tasks show little or even negative gain.
> - **(1b.3)** We have revised the wording in `Section 4.3`, `L425-428`. The updated version highlights that interleaved reasoning is most effective for visual-perception tasks, and clarifies that improvements on physical-world tasks are present but remain modest.
>
> ---
>
> > **Q2(a)**: *It is difficult to infer anything from the % reported in the paper, none of them mention if its absolute, or relative and relative with respect to what.*
>
> **A2(a)**: Thank you for this constructive suggestion!
> We have clarified this in the paper by adding a detailed evaluation protocol (`Section 4.1`, `L368–371`), specifying that all reported percentages are normalized scores on a 0–100 scale based on 5-point GPT-4.1 judgments, with Acc. in ROVER-TG referring to exact answer accuracy (\%) for VQAs.
>
> ---
>
> > **Q2(b)**: *Visual generation performance on pg 5 last paragraph is vague. From reading context, I can map it to RV but would urge the authors to make explicit connections between numbers and metrics, especially when they define them.*
>
> **A2(b)**: Thank you for the constructive feedback.
> - **(2b.1)** We have clarified this by explicitly referring to the metric names in `Section 4.2`, `L394-399`, ensuring a clear connection between the narrative and the reported metrics.
> - **(2b.2)** As stated in the `Table 2` caption, RV is the only metric in `Table 2` that measures visual-generation quality, while RP and Alignment evaluate verbal reasoning and reasoning–visual consistency.

---

> ### Author Response · Authors · 2025-11-17
> **Rebuttal by Authors (2/3)**
>
> > **Q2(c)**: *Section 4.3 Cross-modal Reasoning matters for UMMs: Could not follow through this analysis, CLIP-I and edit world are introduced out of the blue without prior context. Fig on pg 9 top right has the corresponding details but is not reference in text and the figure itself is unclear, with some bars having +ve/-ve value and being of different lengths + no caption. Could not make sense of this at all.*
>
> **A2(c)**: Thank you for the detailed and constructive feedback. We have revised this part of the paper to improve clarity. The updated version in `Section 4.4` (`L464–476`) includes the following changes:
> - **(2c.1)** We now introduce EditWorld, CLIP-I, and CLIP-T in the caption of  `Figure 6` to clarify why EditWorld is used as a conventional editing baseline for contrast.
> - **(2c.2)** The right side of `Figure 6` previously contained incorrect annotations. We have corrected this typo and clarified in the caption that the values denote relative differences between FLUX+GPT vs. FLUX and BAGEL-Think vs. BAGEL.
> - **(2c.3)** We have refined the wording to clearly convey our insight: while external textual refinement can improve conventional editing benchmarks such as EditWorld, **it does not translate to the cross-modal reasoning required in ROVER**.
> - **(2c.4)** We appreciate your feedback and believe these revisions significantly improve the clarity and coherence of our discussion.
>
> ---
>
> > **Q3(a)**: *Human correlation of RV one of the important metrics for IG is low.*
>
> **A3(a)**: Thanks for your valuable comment.
> - **(3a.1)** The lower correlations in our initial RV reliability study were mainly due to limited score variance. Among 200 samples from three models (Nano-Banana, GPT-5, BAGEL-Think), more than 30% outputs fell in the mid-range (3–4). Human experts tended to choose 3, while GPT-4.1 was slightly more lenient and chose 4, which reduces the linear correlation, **even though GPT-4.1 still reliably distinguishes high-quality samples from low-quality ones**.
> - **(3a.2)** GPT-based evaluation has become the **standard and well-validated protocol** in recent visual generation and editing benchmarks (e.g., KRIS-Bench [1], RISEBench [2], WISE [3], IntelligentBench [4], GEditBench [5]), and we adopt the same protocol for comparability and consistency, as it remains the most reliable assessment for complex multimodal generation.
> - **(3a.3)** To further strengthen the reliability analysis, we expanded our study to **8 human judges** for 10 models (1000 instances) with larger score variance and diverse samples, and updated the results in `Figure 8`. RV now achieves **r = 0.75 and MAE = 0.69**, which are comparable to or better than prior benchmarks [1,2]. Importantly, **GPT-based RV scores yield the same model ranking as human scores**, supporting the reliability of our evaluation protocol.
>
> ---
>
> > **Q3(b)**: *Only IG metrics undergo reliability evaluation what about TG metrics which are also llm judges?*
>
> **A3(b)**: We sincerely appreciate this suggestion. We have added TG reliability results in `Appendix C`, showing that GPT-4.1 maintains strong consistency with human experts across both IR and Alignment metrics, with **average correlations around 0.80 and mean absolute errors around 0.56**.
>
> ---
>
> > **Q3(c)**: *The models used for judge calibration are either closed source models or the strongest open-source model. This could potentially add bias in score calibration, having a weaker open source model being part of calibration can ensure that the entire spectrum of scores is calibrated.*
>
> **A3(c)**: Thank you for the helpful suggestion!
> - **(3c.1)** As noted in our earlier response, our evaluation setup follows the common practice in recent image generation and editing benchmarks, where judge calibration is typically performed using GPT-family models.
> - **(3c.2)** To provide a broader coverage of judge capacities, we additionally evaluated two open-source VLM judges of different sizes on 1000 instances across 10 models and report correlation and MAE between human ratings and VLM scores:
>
> | Judge (Average on ROVER-IG)   | Correlation (r) ↑ | MAE ↓ |
> |---------------------------|-------------|--------|
> | **GPT-4.1**               | 0.81        | 0.58   |
> | **Qwen3-VL-32B-Thinking** | 0.74        | 0.68   |
> | **Qwen3-VL-8B-Thinking** | 0.70        | 0.72   |
>
> - **(3c.3)** As shown in the results, all three judges maintain reasonable human-level agreement, further supporting the robustness of our evaluation protocol.

---

> ### Author Response · Authors · 2025-11-17
> **Rebuttal by Authors (3/3)**
>
> > **Q4(a)**: *Fig 4, it would be nice to see the reasoning generated by the models in addition to input text and generated image, to better analyze reasoning alignment.*
>
> **A4(a)**: Thank you for this valuable suggestion! Due to space constraints, the full verbal reasoning sequences for `Figure 4` are now provided in `Figure 10` (`Appendix D`) and referenced in the `Figure 4` caption, providing examples for analyzing reasoning processes more thoroughly.
>
> ---
>
> > **Q4(b)**: *Table 2 does not clearly indicate which models are interleaved vs single turn, image editing only vs UMM but uses these terminologies in the analysis section when Table 2 is referenced. Appendix does provide some insight to them but still models like Show-o2, Blip3o-8b, Janus-Pro-7b, etc have not been classified, making it difficult to relate to the claims in the paper.*
>
> **A4(b)**: Thank you for this constructive comment!
> - **(4b.1)** We have clarified these distinctions in the revised version. Specifically, we updated the caption of `Table 2` to explicitly denote which models are interleaved versus non-interleaved. `Table 4` already separates Image Editing Models and UMMs.
> - **(4b.2)** In addition, we added a complete model summary in `Appendix E`, with a brief description of each model's architecture and how each model is evaluated.
>
> ---
> **References:**
>
> *[1] Wu et al. KRIS-Bench: Benchmarking Next-Level Intelligent Image Editing Models. NeurIPS 2025.*
>
> *[2] Zhao et al. Envisioning Beyond the Pixels: Benchmarking Reasoning-Informed Visual Editing. NeurIPS 2025.*
>
> *[3] Niu et al. WISE: A World Knowledge-Informed Semantic Evaluation for Text-to-Image Generation. arxiv 2025.*
>
> *[4] Deng et al. Emerging Properties in Unified Multimodal Pretraining. arxiv 2025.*
>
> *[5] Liu et al. Step1x-edit: A Practical Framework for General Image Editing. arXiv 2025*
>
> ---
>
> Thank you for your valuable comments. We have revised our manuscript based on your suggestions and are committed to further refining the paper to improve its readability. We sincerely appreciate your help once again!

---

> > ### Comment · Reviewer_FBtf · 2025-11-19
> >
> > I appreciate the feedback on the comments and the authors’ efforts to improve the paper’s clarity. I am raising my score to 6.

---

> > > ### Author Response · Authors · 2025-11-19
> > > **Thank you for your acknowledgment!**
> > >
> > > Dear Reviewer `FBtf`,
> > >
> > > Thank you for your positive response to our rebuttal and for informing us of your plan to vote for acceptance.
> > >
> > > We are delighted that our responses have addressed your concerns, and we truly appreciate your valuable feedback and encouragement.
> > >
> > > Best regards,
> > >
> > > The Authors of Submission 1565

---

### Official Review · Reviewer_Apxs · 2025-11-01

**Soundness:** 3
**Presentation:** 3
**Contribution:** 3
**Rating:** 6
**Confidence:** 4

**Summary:**

The paper introduces ROVER, the first human-annotated benchmark explicitly designed to evaluate reciprocal cross-modal reasoning in Unified Multimodal Models (UMMs). It addresses the fundamental limitation of existing benchmarks, which treat understanding and generation abilities in isolation, failing to assess how one modality can guide, verify, or refine outputs in the other. 17 SOTA UMMs have been evaluated on the visual generation and text generation settings.

**Strengths:**

- The paper is well structured and presented overall, with a helpful project page.
- It addresses an important gap in UMMs by benchmarking and evaluating reciprocal cross-modal reasoning.

**Weaknesses:**

- Table 1 should include comparisons across more aspects. Additional explanations are needed in both the text and the table caption: benchmark dataset scale, whether it is for VG/TG/both, and clarifications on the multi-dimensional and hybrid evaluations and the types.
- This work's emphasis on intermediate reasoning as a core signal for multimodal reasoning distinguishes it from existing benchmarks. However, the data curation process for these progressive reasoning steps is under-specified, especially for the TG setup. The paper should clarify exactly what intermediate data is curated for various sub-tasks, dataset statistics, and how it is used for evaluation.
- The current evaluation depends only on GPT-based judgment. Introducing objective, automatically computed metrics would improve the reliability of the fine-grained reasoning evaluation.
- Would it also be beneficial to include the text reasoning chain for the TG task? Clarification is needed on how the progressive visual reasoning steps are validated as active reasoning components rather than decorative elements, as claimed in line 246.

**Questions:**

Please see the weakness above.

---

> ### Author Response · Authors · 2025-11-17
> **Rebuttal by Authors (1/2)**
>
> We sincerely appreciate your constructive comments and your recognition of our paper’s clarity and contribution. We will explain your concerns point by point.
>
> ---
>
> > `Q1`:  *Table 1 should include comparisons across more aspects. Additional explanations are needed in both the text and the table caption.*
>
> **A1**: Thank you for your insightful suggestions!
> - **(1.1)** We have revised `Table 1` accordingly and added the requested clarifications for each column.
> - **(1.2)** As illustrated in `L148–161`, all benchmarks compared in `Table 1` are visual-generation benchmarks, so distinguishing IG/TG/both is not applicable. Additional discussion of interleaved reasoning is provided in `Appendix C`.
>
> ---
>
> > `Q2`:  *The data curation process for these progressive reasoning steps is under-specified, especially for the TG setup. The paper should clarify exactly what intermediate data is curated for various sub-tasks, dataset statistics, and how it is used for evaluation.*
>
> **A2**: We sincerely appreciate your question. In the revised version, we have added detailed clarifications for the reasoning step data.
> - **(2.1)** For ROVER-IG tasks, we do not provide ground-truth verbal reasoning steps; these tasks are evaluated based on the input and final visual outputs. Evaluation prompts are included in `Appendix E.1`.
> - **(2.2)** For ROVER-TG tasks, we provide visual intermediate reasoning images for most tasks, including:
>     - human-annotated geometric auxiliary lines for logical tasks;
>     - robot-arm trajectories and physical simulator frames extracted from original video data;
>     - full target images in jigsaw for visual-perception tasks
> - **(2.3)** We add clarifications in `L301-304` and `Appendix B`, which include data curation procedure, task-level statistics and explain how these visual references are incorporated into TG evaluation.
>
> ---
>
> > `Q3`:  *The current evaluation depends only on GPT-based judgment. Introducing objective, automatically computed metrics would improve the reliability of the fine-grained reasoning evaluation.*
>
> **A3**: Thank you for the constructive suggestion!
> - **(3.1)**  GPT-based evaluation is the **standard and well-validated protocol** in recent visual generation and editing benchmarks (e.g., KRIS-Bench [1], RISEBench [2], WISE [3], IntelligentBench [4], GEditBench [6]), and we adopt the same protocol for comparability and consistency, as it remains the most reliable assessment for complex multimodal generation.
> - **(3.2)**  Existing automatically computed visual metrics (such as GenEval [5]) **often exhibit specific biases**. For example, GenEval cannot detect complex and uncommon objects or phenomena, such as clothing related to a particular culture or the morphology of plant veins [7].
> - **(3.3)**  Following the reviewer’s suggestion, we **additionally incorporate GenEval [5] as an objective metric** alongside GPT-4.1 judgments for 1000 instances from 10 unified models . As shown below, GPT-4.1 results on Reasoning Visual (RV) metric aligns substantially better with human evaluations models:
> | Judge       | Correlation (r) with human |
> |--------------|------------------------|
> | GPT-4.1 (Ours) |  0.75  |
> | GenEval |      0.56 |
> These results suggest that **GPT-based evaluation more faithfully** captures the multimodal reasoning quality that ROVER aims to assess. We agree that developing more robust automatic evaluation metrics is a valuable future direction, and we appreciate the reviewer for raising this point.
>
> ---
>
>
> **Reference**:
>
> *[1] Wu et al. KRIS-Bench: Benchmarking Next-Level Intelligent Image Editing Models. NeurIPS 2025.*
>
> *[2] Zhao et al. Envisioning Beyond the Pixels: Benchmarking Reasoning-Informed Visual Editing. NeurIPS 2025.*
>
> *[3] Niu et al. WISE: A World Knowledge-Informed Semantic Evaluation for Text-to-Image Generation. arxiv 2025.*
>
> *[4] Deng et al. Emerging Properties in Unified Multimodal Pretraining. arxiv 2025.*
>
> *[5] Ghosh et al. GenEval: An Object-Focused Framework for Evaluating Text-to-Image Alignment. ICLR 2024.*
>
> *[6] Liu et al. Step1x-edit: A Practical Framework for General Image Editing. arXiv 2025.*
>
> *[7] Ye et al. Echo-4o: Harnessing the Power of GPT-4o Synthetic Images for Improved Image Generation. arXiv, 2025.*

---

> ### Author Response · Authors · 2025-11-17
> **Rebuttal by Authors (2/2)**
>
> > `Q4`:  *Would it also be beneficial to include the text reasoning chain for the TG task? Clarification is needed on how the progressive visual reasoning steps are validated as active reasoning components rather than decorative elements.*
>
> **A4**: Thank you very much for your insightful question!
> - **(4.1)**  For TG tasks, interleaved models naturally produce a text-image reasoning chain as part of their output, and our evaluation metrics already consider interleaved text-image reasoning as shown in prompts in `Appendix E.1`.
> - **(4.2)**  We have added a more detailed description of the data curation process for TG in `Appendix B` for further clarification. Specifically,
>
>    - For logical problems, we curated over 1,000 instances with ground-truth visual annotations. We performed an automatic sanity check using GPT-5 and selected 150 cases where the presence of the ground-truth visual CoT changes the model’s answer, indicating that **visual reasoning steps have a concrete impact on model behavior**.
>    -  For visual and physical reasoning tasks, our task definitions inherently require intermediate visual cues. Examples include robot arm trajectories in manipulation tasks, complete reference images in jigsaw tasks, and wide-view images that support spatial understanding in multi-view spatial reasoning. These intermediate visual cues encode fine-grained geometric and perceptual structure that **is hard to capture through text-only reasoning**.
> ---
> Thank you once again for Reviewer `Apxs`'s recognition and constructive suggestions, which have been instrumental in enhancing the quality of our research!

---

> ### Author Response · Authors · 2025-11-26
> **Follow-up on Rebuttal: Discussion Phase Closing Soon**
>
> Dear Reviewer `Apxs`,
>
> We sincerely appreciate your invaluable feedback, which has significantly contributed to the improvement of our work.
>
> Following your suggestions, we revised `Table 1` to improve comparisons and detailed the data curation for intermediate reasoning steps in `Appendix B`. We also validated our evaluation protocol by incorporating GenEval, demonstrating that our metric aligns better with human judgment. Finally, we clarified the text-image reasoning process and provided evidence verifying the active contribution of visual steps to model performance.
>
> We hope that our revisions and clarifications have resolved your concerns. If you find our response satisfactory, we would be deeply grateful for a reconsideration of our score. Otherwise, if you have any additional questions, please do not hesitate to let us know. We would be more than willing to provide further clarification.
>
> We are truly grateful for your insightful comments, which have helped us improve the clarity and completeness of our work!
>
> Best regards,
>
> The Authors of Submission 1795

---

### Author Response · Authors · 2025-11-17
**General Response**

**Dear Reviewers, ACs, and SACs,**

We deeply appreciate the insightful and valuable comments provided by all reviewers.

---

We are grateful for the reviewers' recognition of this work as an important step toward evaluating **reciprocal cross-modal reasoning in unified multimodal models**. ROVER provides a carefully designed benchmark that captures how models integrate visual and verbal reasoning, and our analyses reveal key strengths and limitations of current UMMs. We believe ROVER provides a rigorous foundation for advancing cross-modal reasoning in next-generation omnimodal models.

Overall, we are encouraged by the reviewers' positive feedback, which highlights:
- The **motivation and novelty** of reciprocal cross-modal reasoning are clear and impactful (Reviewers `Bjyh`, `FBtf`, `Apxs`).
- The benchmark is **comprehensive, well-structured, and carefully designed** across domains and reasoning types (Reviewers `FBtf`, `Apxs`).
- The **experiments** and hybrid VLM–expert evaluation are thorough and **clearly presented** (Reviewer `icDF`).
- The **findings** on interleaved generation, reciprocal modality **benefits**, and reasoning gaps are **insightful** and valuable for future UMM development (Reviewers `Bjyh`, `icDF`, `FBtf`).

---

To address the reviewers' main concerns, we have conducted several additional experiments and analyses, including:
- **Added GenEval evaluation** to compare automatic objective metrics with GPT-4.1 (Reviewer `Apxs`).
- **Expanded the human–VLM reliability study** to 8 human experts, 10 models, and 1000 samples, updating correlation and MAE for GPT-4.1 judge (Reviewers `Bjyh`, `FBtf`).
- **Incorporated open-source VLM judges** to improve calibration robustness and explore more cost-efficient evaluation options (Reviewers `Bjyh`, `FBtf`).
- **Clarified the necessity and active utility of visual reasoning steps** by refining data curation descriptions  (Reviewers `Apxs`, `icDF`).
---

**Summary of revisions:**
- Clarified benchmark data curation and evaluation setup in `Section 3.2` and `Appendix B`, `E.1`
- Added TG reliability results in `Appendix C` and updated `Figure 8` with an extended human validation study
- Refined analysis in `Sections 4.2`, `4.3`, and `4.4` to present the results and insights more clearly and accurately.
- Updated `Table 1`, `2` and `Figures 4`, `6` to clarify metrics, model types, and annotations.
- Revised `Section 4.1` with more experimental setup details.
- Expanded verbal reasoning examples for visual generation in `Appendix D`.

All revisions in the paper are highlighted in blue. We sincerely appreciate the reviewers' constructive suggestions and remain committed to continually improving our work.

---

We address each reviewer's comments point by point below. We welcome further discussion and look forward to continued engagement. Thank you!

---

### Author Response · Authors · 2025-11-28
**Summary of Rebuttal and Final Remarks**

**Dear Reviewers, ACs, and SACs,**

We sincerely appreciate the constructive and thorough feedback provided by the reviewers. We are pleased that the main concerns raised by the reviewers have been successfully addressed through the extensive additional experiments, clarifications, and revisions conducted during the rebuttal.

As summarized in our `General Response`, we received positive comments in the initial reviews reflecting the reviewers' recognition of our work's motivations and contributions. Following the discussions with the reviewers, our scores improved to **(6, 6, 6, 6)**. Specifically, **Reviewer `Bjyh` and Reviewer `FBtf` raised their scores from 4 to 6 on Nov 17 and Nov 19, respectively, acknowledging that their concerns have been fully addressed and providing more positive evaluations**. We sincerely thank the reviewers for their timely engagement and encouraging support.

While it is deeply regrettable that the OpenReview incident on November 27 prevented further engagement, we remain grateful for the insightful discussions prior to the disruption, which have helped further solidify our contributions and ultimately embodies the true spirit of the ICLR discussion process.

---

We believe this work offers distinct contributions to the **Unified Multimodal Model** and **multimodal reasoning** community:

* **ROVER bridges the gap between isolated unimodal evaluations and true omnimodal generation by establishing a benchmark for reciprocal cross-modal reasoning.** It highlights that actively using one modality to guide, verify, or refine the other is a critical frontier for the next generation of unified multimodal models.

* **Our evaluation offers key insights into the current landscape of unified models, revealing significant deficiencies in cross-modal reasoning.** These findings provide a clear direction for future improvements.

* To foster future research and ensure reproducibility, **we are committed to releasing all datasets and inference and evaluation code.  We aim to provide a transparent and reproducible pathway for the community to build upon our findings in ROVER**.

---

All revisions are marked in `blue` in the updated manuscript and will be included in the camera-ready version. **For a more detailed summary of the rebuttal and revisions, please refer to our `General Response`.**

Thank you once again for your time, expertise, and invaluable service to the research community.

With sincere appreciation,

The Authors of Submission 1795

---

### Meta-Review · Area_Chair_fSwf · 2026-01-06

**Summary:**

This paper introduces ROVER, a new benchmark designed to evaluate reciprocal cross-modal reasoning, meaning whether multimodal models can actively use text to guide image generation and images to support textual reasoning. Reviewers generally agreed that this fills a real gap in existing evaluations, which often treat vision and language in isolation. AC found that the benchmark is carefully constructed, covers a wide range of tasks, and the large-scale evaluation across many state-of-the-art models reveals consistent and useful patterns about current model limitations.

**Reviewer Concerns:**

Several concerns were raised during review. One recurring issue is the heavy reliance on VLM-as-judge evaluation. While the authors provided additional human validation and open-source judge comparisons in the rebuttal, some reviewers still felt uneasy about judging fine-grained reasoning quality through another large model, especially given the remaining gap between human and automated scores. This does not invalidate the benchmark, but it does mean that conclusions about “reasoning quality” should be interpreted with lots of caution.

Another concern is that, in some cases, language-only models perform competitively with unified multimodal models, which raises questions about how strongly certain tasks truly require cross-modal reasoning. The authors clarified that this reflects current model weaknesses rather than task design flaws, and provided evidence that intermediate visual reasoning steps can change model behavior. Still, this ambiguity limits how sharply the benchmark can separate multimodal reasoning from strong language priors.

Also, reviewers pointed out over-claims and presentation issues in the original draft, including unclear metric definitions and analysis sections that were hard to follow. These issues were largely addressed in the rebuttal through clearer wording, revised figures, and expanded explanations and AC confirms.

Overall, AC recommend acceptance, with the understanding that the benchmark’s conclusions should be read as diagnostic for the community to conduct futher research and validation.

**Reviewer Scores:**

FBtf to 6
Bjyh to 6

So final 6 6 6 6

---

### Decision · Program_Chairs · 2026-01-26

Accept (Poster)